# Test-Time Training Done Right

## Abstract

Test-Time Training (TTT) models context dependencies by adapting part of the model's weights (often referred to as fast weights) at inference time. This adapted fast weight, similar to recurrent states in RNNs, stores temporary memories of past tokens in the current sequence. Existing TTT methods have struggled to demonstrate effectiveness in handling long-sequence data, due to their computational inefficiency on modern GPUs. The TTT layers in many of these approaches operate with extremely low FLOPs utilization (often below 5%) because they deliberately apply small online mini-batch sizes (e.g., updating fast weights every 16 or 64 tokens). Moreover, a small mini-batch implies fine-grained block-wise causal dependencies in the data, making them unsuitable for data beyond 1D ordered sequences, like sets or N-dimensional grids such as images or videos. In contrast, we pursue the opposite direction by proposing an extremely large chunk update, ranging from 2K to 1M tokens across tasks of varying modalities, which we refer to as Large Chunk Test-Time Training (LaCT). This approach improves hardware utilization by orders of magnitude, and more importantly, facilitates scaling of non-linear state size (up to 40% of model parameter size), hence substantially improving state capacity, all without requiring cumbersome and error-prone custom kernel implementations. It also allows easy integration of sophisticated optimizers like Muon for online memory updates. We validate our approach across diverse data modalities and tasks, including novel view synthesis from image sets, language models, and auto-regressive video diffusion models. Our approach can scale up to 14-billion-parameter auto-regressive video diffusion models handling sequences of up to 56K tokens. In our longest sequence experiment, we perform novel view synthesis with more than one million context length. Our results highlight the computational and performance benefits of large-chunk test-time training, paving the way for more efficient and scalable long-context sequence modeling. We hope that this work will inspire and accelerate new research in the field of long-context modeling and test-time training.

## 1 Introduction

The demand for handling long contexts is rapidly growing. While softmax attention [1] has become the de facto solution for modeling various types of data, its computational cost grows quadratically with sequence length, motivating extensive research into more efficient long-context modeling.

Recently, Test-Time Training (TTT) [2] has emerged as a promising approach for efficient sub-quadratic sequence modeling. TTT extends the concept of recurrent states in RNNs to a small, online-adapted sub-network. The parameters of this sub-network also referred to as fast weight [3], as they are rapidly adapted online via self-supervised objectives to memorize in-context information. In other words, the context is compressed into the finite-size fast weights, allowing for efficient sequence

processing. Numerous recent studies [4, 5, 6, 7] have explored various online objectives, optimizers, and architectures for fast weight networks.

Despite these efforts, existing TTT methods struggle to scale effectively to long contexts, primarily due to extremely low hardware utilization in their TTT layers (often below 5% peak FLOPS on modern GPUs). This inefficiency is because of the usage of small mini-batch sizes, i.e. updating fast weights every token or every 16 to 64 tokens, which is conventionally assumed to be more effective for in-context learning. Such small mini-batch results in poor parallelism and low compute intensity, and presents significant challenges for hardware-efficient implementation, especially when using large, nonlinear fast weights, making it difficult to achieve non-trivial (above 10%) FLOPs utilization.

In this paper, we adopt the opposite strategy and introduce Large Chunk Test-Time Training (LaCT). LaCT leverages extremely large chunk (from 2048 to 1M tokens) as the basic unit to update the fast weight. Since the tokens within each large chunk are treated as an unordered set, we further integrate window attention into LaCT to capture local dependencies within the chunk. LaCT significantly enhances parallelism, leading to substantially improved GPU utilization (up to 70% on NVIDIA A100s) with just a few dozen lines of pure PyTorch code (see the Appendix). This efficiency enables the scaling of non-linear fast weights to enhance the memory capacity. And simple implementation allows easy integration of more effective test-time optimizers, such as Muon [8]. Furthermore, LaCT's large-chunk design is also natural to model diverse N-dimensional data as we can align chunk-size with the internal structure of the data (e.g., grouping tokens within an image or consecutive video frames as a chunk).

We extensively validate LaCT on three tasks spanning different modalities and data structures:

- *Novel View Synthesis.* Our model is capable of processing up to $128$ input images at a resolution of $960 \times 536$ leading to a maximum of 1M tokens, and outperforms 3D Gaussian Splatting [9] in terms of rendering quality under such input scale.

- *Language Modeling.* Our model achieves competitive performance compared to SoTA methods such as DeltaNet [10], even though a chunk structure is not explicitly present.

- *Autoregressive Video Diffusion.* We adapt a 14-billion-parameter bidirectional video diffusion transformer into an autoregressive model by incorporating LaCT with sliding window attention. This adapted model generates consistent videos up to 56,000 visual tokens.

To summarize, our approach establishes an efficient, scalable, and highly performant framework for long sequence modeling across diverse modalities. By removing the dependency on low-level, hardware-specific implementations, LaCT enables broader exploration of the architectural design space. We believe this can democratize research in efficient long-context modeling and inspire the development of more novel and effective designs.

## 2 Preliminary

### 2.1 Test-Time Training

Consider a one-dimensional sequence of $N$ tokens $\mathbf{x} = [x_1, x_2, \ldots, x_N]$, where each token $x_i \in \mathbb{R}^d$. Following attention formulation, each input tokens $x_i$ is projected into query ($q_i$), key ($k_i$), and value ($v_i$) vectors. For clarity, we assume all these vectors $q_i, k_i, v_i \in \mathbb{R}^d$.

Test-Time Training (TTT) [2] introduces a neural network with rapidly adaptable weights—called *fast weights* [3]—that are updated during both training and inference to dynamically store context information. This contrasts with the *slow weights* (i.e., model parameters) that are frozen during inference. Formally, TTT defines fast weights in the form of a neural network: $f_W(\cdot) : \mathbb{R}^d \to \mathbb{R}^d$ parameterized by the fast weights $W$, and it involves two primary operations:

$$\textbf{Update operation:} \quad W \leftarrow W - \eta \nabla_W \mathcal{L}\big(f_W(k), v\big) \tag{1}$$

where $\mathcal{L}(\cdot, \cdot)$ is a loss function between the transformed key $f_W(k)$ and the value $v$, commonly Mean Squared Error, designed to encourage the network to associate keys with corresponding values. $\eta$ is the learining rate. Intuitively, this learning objective is to encode the KV cache into a neural memory with fixed state size as *accurate* as possible [4].

$$\textbf{Apply operation:} \qquad o = f_W(q), \tag{2}$$

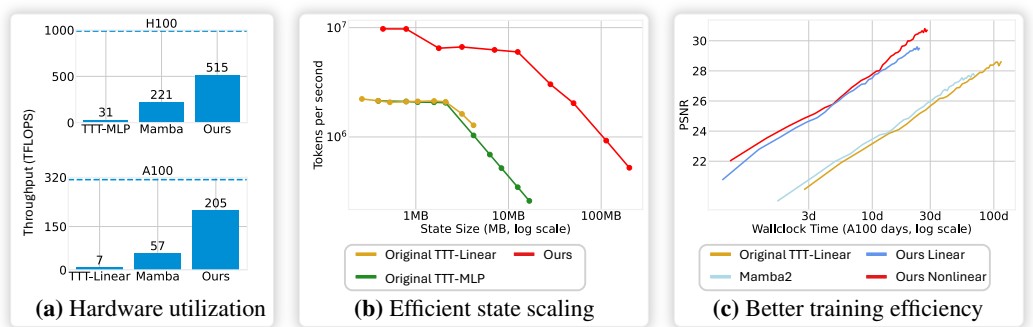

| (a) Hardware utilization | (b) Efficient state scaling | (c) Better training efficiency |

Figure 1: Using larger chunk sizes significantly improves GPU utilization compared to the original test-time training (TTT) method that even uses customized kernels **(a)**. This enhanced utilization enables efficient scaling to larger state sizes **(b)**, resulting in improved training efficiency and better overall performance **(c)**. The dotted line in **(a)** is the theoretical peak BF16 throughput of the GPU.

where the updated fast weights $W$ are used to compute the output vector $o$ given the query $q$. The per-token TTT layer iteratively perform the update and apply operations on each token $x_i$ in sequence.

## 2.2 Challenges in Efficient Implementation

Frequent online update of fast weights is inefficient due to memory bandwidth limitations. Consequently, previous works [11, 12, 13, 14, 15] often employ customized kernels that keep fast weights in Streaming Multiprocessor (SM) memory across updates to reduce memory load. However, this strategy typically requires fast weights to evolve mostly independently within SMs to reduce communications, which is not valid for large nonlinear states (e.g., the nonlinear SwiGLU fast weight in Sect. 3.1 and the Muon update in Sec. 3.2). Moreover, developing such kernel code is cumbersome, with far longer development cycles than native PyTorch code, hindering rapid research exploration.

On the other hand, a PyTorch-based implementation, while simpler, is typically bounded by memory speed. As an illustration, consider a PyTorch implementation of simple MLP fast weight, the core of which is a matrix multiplication between fast weight (e.g., $h \times h$ matrix) and the mini-batch input ($b \times h$ where b is the chunk size). The ideal compute-to-memory ratio is:

$$r = \frac{2h^2 b}{2h^2 + 4hb} = \frac{h/2}{1 + \frac{h}{2b}} = \frac{b}{1 + \frac{2b}{h}} \leq \min(h/2, b) \tag{3}$$

Here, $2h^2 b$ is the FLOPs to for matrix multiplication, the denominator $2h^2 + 4hb$ is the memory workload for two input matrices and the output in BF16 (2 bytes). Small fast weight size (e.g., $h = 64$) or small chunk size (e.g., $b = 16$) will bound the ratio $r$ far below the theoretical peak (e.g., 290 FLOPs per byte on H100), making the operation memory-bound and limiting compute usage.

In light of this, we advocate for using large chunk sizes (from $2048$ to $1M$). This allows us to achieve higher throughput (Fig. 1a) with better training efficiency and performance (Fig. 1c). Our design also allows the state size to be scaled up efficiently (Fig. 1b), leading to significant results improvement with such scaling (Fig. 7a). Our architecture achieves a state-to-parameter size ratio $\geq 40\%$, which is an order of magnitude larger than previous methods' ratio of $0.1\%$ to $5\%$.

## 3 LaCT Model Architecture

As shown in Fig. 2, LaCT block consists of three types of layers: a window attention layer, a large-chunk TTT layer, and a feed-forward layer. Each layer is equipped with residual connections [16] following the practice in Transformer [1]. The window attention layer performs local self-attention to capture the local dependency. In the TTT layer, we split the sequence into large chunks. The history context is gradually compressed into the fast weights through an 'update' operation (regarding key vectors $K$ and value $V$), and latest weight is 'applied' to the current query vector (Q) for computing its corresponding output. The feed-forward layer performs channel mixing as in Transformer. We omit several linear and normalization layers in Fig. 2 for clarity and details are in Appendix. Our framework offer great flexibility in handling diverse data types. In this section, we present the general designs in our approach and later describe data-specific variations in Sec. 4.

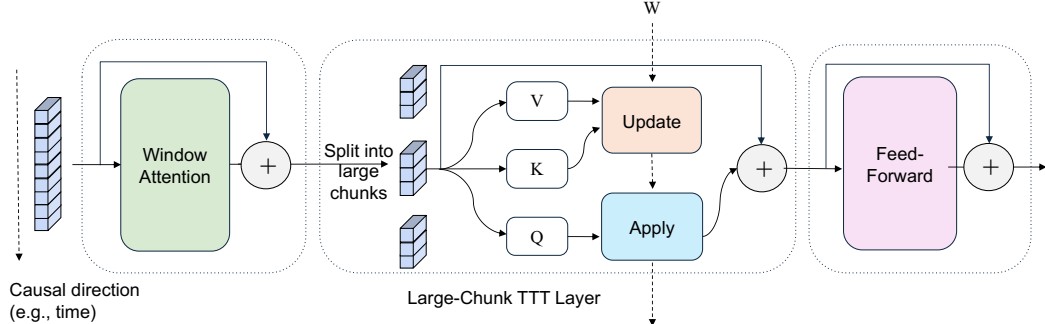

Figure 2: The basic diagram for a LaCT block. The large-chunk TTT layer updates the fast weight $W$ to store history chunk information, while the window attention handles the internal structures within the chunk. The solid line denotes the information flow over model depth and the dashed line denotes the information flow over time (i.e., the fast weight $W$ passing through chunks). Instantiations in Sec. 4 use different chunk sizes and window attention types according to the specific data structure.

### 3.1 Large-Chunk TTT Layer

Different from the per-token update in Eqn. 1, the chunk-wise update computes the gradient of the summed loss over all keys $\{k_i\}$ and values $\{v_i\}$ within the chunk. As the chunk size is large, weight updates are performed infrequently. This enables more sophisticated weight-update rule designs (discussed in Sec. 3.2) and amortizes the update cost. The 'update' operation for the fast weight is:

$$g = \nabla_W \sum_{i=1}^{b} \eta_i \mathcal{L}\big(f_W(k_i), v_i\big) \tag{4}$$

$$W \leftarrow \text{weight-update}(W, g), \tag{5}$$

where $b$ is the chunk size, $g$ is the gradient of the fast-weight loss function, and $\eta_i$ is the learning rate of each token (usually predicted from input tokens). The 'apply' operation $o_i = f_W(q_i)$ is the same as Eqn. 2 and all query vectors $\{q_i\}$ in the chunk share the same updated fast weight $W$.

Motivated by recent LLMs [17], we adopt SwiGLU-MLP [18] without bias terms as the fast-weight network. Our fast weights consists of three weight matrix $W = \{W_1, W_2, W_3\}$, and the network is:

$$f_W(x) = W_2 \left[\text{SiLU}(W_1 x) \circ (W_3 x)\right] \tag{6}$$

where $\circ$ is an elementwise multiplication. We apply a simple dot product loss as our loss function:

$$\mathcal{L}\big(f_W(k_i), v_i\big) = -f_W(k_i)^\top v_i \tag{7}$$

**Execution orders for 'apply' and 'update'.** Note that the 'update' operation and 'apply' operation of TTT are decoupled, and we can set the chunk size adaptively and apply these operation in different orders; this allows us to model diverse kinds of data dependencies, similar to different attention masks in self-attention. Figure 3 illustrates this concept. In Figure 3a, when the chunk size equals the full sequence length, performing the apply followed by the update operation is conceptually similar to full attention. Using update and apply alternately leads to a block-wise causal mask (Fig. 3b), where the block size corresponds to the chunk size. Switching the order between the two operations results in the a shift in the mask (Fig. 3c). This shifted mask does not leak future information within the chunk and is important when building the full causal mask in Language Modeling (Sec. 4.2). Moreover, only updating on a subset of chunks and applying to all (Figure 3d) is analogous to strided block-wise causal mask.

### 3.2 Non-Linear Update of Fast-Weight

Fast-weight updates in TTT repeatedly accumulate gradients, and thus suffer from magnitude explosion or decayed memory. Large-chunk TTT allows non-linear updates to improve stability and effectiveness while preserving efficiency. For the 'weight-update' operation in Eqn. 5, our vanilla implementation involves gradient descent followed by weight normalization:

$$\text{weight-update}(W, g) = \text{L2-Normalize}(W - g). \tag{8}$$

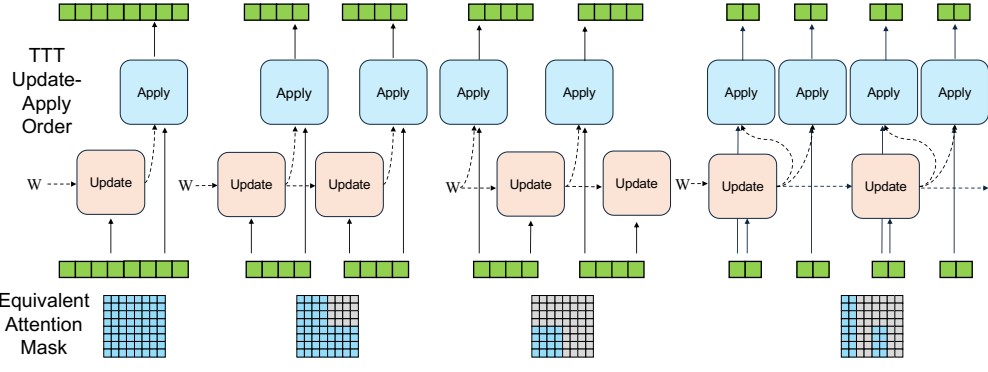

Figure 3: Different 'Update' and 'Apply' orders and their equivalent attention mask.

We have also explored a more robust nonlinear Muon [8] update rule [1] with weight normalization:

$$\text{weight-update}(W, g) = \text{L2-Normalize}(W - \text{Muon}(g)) \tag{9}$$

**Fast-weight normalization.** We apply L2 weight normalization [19] to the updated fast weights along the input dimension. We do not use explicit weight-decay term as in previous methods [5, 20, 13, 11]. When the network is conceptually rotated 90 degrees, treating the sequence dimension as the depth of a virtual model, the test-time training updates act as residuals over time [16]. In this view, our fast-weight normalization is analogous to the *post-layer norm* in Transformer architectures, which constrains activation scales within the residual path.

**Muon-update rule.** Essentially, Muon normalizes the spectral norm of matrix gradient using Newton-Schulz iterations. In short, let $g = USV^T$ be the Singular Value Decomposition(SVD) of the gradient $g$, then Muon operator approximately converts the gradient as:

$$\text{Muon}(g) \simeq UV^T \tag{10}$$

Muon also improves the numerical stability in our setup. For example, the learning rate ($\eta_i$ in Eqn. 4) now only reflects the relative importance of tokens within a chunk as Muon normalizes the absolute scale. See [8] and Appendix for analysis of its computational cost.

### 3.3 Window Attention

Large-chunk TTT layer models data as a sequence of sets. However, many data modalities are not naturally in such a form, and are instead sequences of grids (e.g., videos), sets of grids (e.g., image collections), or simple one-dimensional sequences (e.g., text). For such modalities, the dependencies within each chunk still matter to capture the structure of the data, thus we apply local window attention—either causal or bidirectional—before TTT layers. Hence, LaCT is a hybrid architecture with the quadratic-compute attention for local structure and linear-compute TTT for long context.

### 3.4 Context Parallelism

Context Parallelism (CP) partitions the sequence along the context length dimension and distributes the shards across multiple devices for parallel computing. The feed-forward layer and window attention are local operators thus natively support CP. For TTT layer, small chunks hardly support CP thus tensor parallelism (i.e., parallel over the heads) is preferred. Our large-chunk TTT layer allows CP by sharding the tokens within a chunk. Suppose each shard contains $s$ tokens, the fast weight gradient of the chunk is the sum over all shard's gradients given the linearity of the gradients:

$$g = \nabla_W \sum_{j=1}^{\text{shards}} \sum_{i=1}^{s} \eta_i \mathcal{L}_i = \sum_{j=1}^{\text{shards}} \nabla_W \sum_{i=1}^{s} \eta_i \mathcal{L}_i \tag{11}$$

---

[1]Muon requires weights in matrix form, and our current fast-weight function SwiGLU-MLP has three matrices as the weights (i.e., $W_1, W_2, W_3$ in Eqn. 6).

This can be implemented through distributed all-reduce-sum and is logically the same as Distributed Data Parallelism (DDP), except that the parameters are the fast weights and input data are the tokens in the chunk. We adopt such parallelism in training the novel view synthesis task (see Sec. 4.1) and observe minimal throughput overheads (1% to 3%). LaCT architecture is compatible with other parallelism strategies (e.g., data parallelism, pipeline parallelism, and tensor parallelism).

# 4 LaCT for N-Dimensional Data

In this section, we introduce the three tasks we address using LaCT—novel view synthesis, language modeling, and autoregressive video generation. These tasks have different inherent data structures and we address them with corresponding design choices.

## 4.1 Novel View Synthesis - Image Set

Novel view synthesis (NVS)[21, 22] aims to render images of a static scene from previously unseen viewpoints. Formally, given a set of $N$ input posed images $\{(I_i, P_i)\}_{i=1}^N$ of a static scene, where $I_i \in \mathbb{R}^{H \times W \times 3}$ is an RGB image and $P_i$ is its corresponding camera pose, the model needs to synthesize new images from novel camera poses that typically do not overlap with the input views.

We find that NVS is an effective test bench for evaluating a model's online memory and compression capabilities. Firstly, NVS is challenging as it requires spatial compression, dense retrieval, and basic physical reasoning. Secondly, NVS can be formulated as a non-generative task, significantly reducing training computation and the need for extensive model parameters to store world knowledge, thereby enabling rapid experimentation. Thirdly, the substantial redundant information in dense input views incentivizes the model to learn effective compressions. Given these observations, we use NVS for our initial research iterations. We find that some of the insights gained are transferrable to other tasks.

Our NVS model follows the basic LaCT diagram in Sec. 3. Both the posed input images and poses of the target novel views are tokenized by patchify and linear layers, following LVSM [23]. The window attention exactly covers the tokens from a single image. The LaCT layer adapts a single-round of strided block-wise causal mask (Fig. 3d), which updates the fast weight using all input image tokens, and applies to both the input and target tokens. The *update* step resembles a prefill stage, while the *apply* operation resembles parallel decoding. During rendering of novel views, each test-time training layer functions as a static weight layer, making the entire model a static vision transformer [24].

## 4.2 Language Modeling - Text Sequence

Autoregressive language models predict the probability distribution of the next token given preceding tokens, $p_\theta(x_n|x_1, \ldots, x_{n-1})$. Text sequences lack inherent chunk structures, so for LaCT, we define chunk size as a hyperparameter (e.g., 2048 or 4096 tokens). We utilize the shifted block-wise causal mask as in Fig. 3(c) for the TTT apply-update sequence to avoid seeing future tokens in a chunk. Since LaCT lacks per-token causality within each chunk, we employ sliding window attention—with window size equal to the chunk size—to efficiently model per-token causal dependencies.

## 4.3 Autoregressive Video Diffusion - Image Sequences

Chunkwise autoregressive video diffusion iteratively denoises a number of subsequent video frames, conditioned on the previously generated clean frames, where each chunk can contain thousands of visual tokens. We use teacher-forcing training by interleaving noisy and clean frame chunks. Specifically, a video of N frame chunks is structured as:

$$S = [X_1^{\text{noise}}, X_1, X_2^{\text{noise}}, X_2, \ldots, X_N^{\text{noise}}] \tag{12}$$

where each noisy chunk $X_i^{\text{noise}}$ is produced by adding unit Gaussian noise $\epsilon$ to the $i$-th clean video chunk as $X_i^{\text{noise}} = X_i(1 - t_i) + \epsilon t_i$ and $t_i \in [0, 1]$ denotes the strength of chunk-independent noise.

To handle such a data structure, we employ the strided block-wise causal mask in Fig. 3d for LaCT. Specifically, it *applies* fast weights to each chunk sequentially while only *updating* fast weights on clean chunks. This simple strategy ensures that each denoising operation only accesses previously cleaned frames. The windowed attention uses a non-overlapping window with 2 consecutive chunks

Table 1: Summary of our experiments on three different data structures. 'd' denotes model dimension.

| Task name | Data Structure | Chunk Size | State Size | Model Size | Max Length | Context Parallelism |
|---|---|---|---|---|---|---|
| Novel View Synthesis | Image set | Full sequence | $6d^2$ | 0.3B | 1M | Within-chunk parallel |
| AR Video Diffusion | Image sequence | Three frames | $3d^2$ | 1.3B, 14B | 56160 | Head-dim parallel |
| Language Models | 1D Sequence | 2K, 4K tokens | $0.75d^2$ | 0.7B, 3B | 32768 | N/A |

Table 2: Complexities of methods on novel view synthesis w/ $n$ input. Prefill and rendering speed are measured on A100 with 48 512×512 input images (196K input tokens, 4K decoding tokens).

| | State Size | Prefill Compute | Decoding Compute | # Params | Prefill speed | Rendering FPS |
|---|---|---|---|---|---|---|
| Full attention | $O(n)$ | $O(n^2)$ | $O(n)$ | 284M | 16.1 s | 2.3 FPS |
| Perceiver Attention | $O(1)$ | $O(n^2)$ | $O(1)$ | 287M | 16.8 s | 34.4 FPS |
| Ours | $O(1)$ | $O(n)$ | $O(1)$ | 312M | 1.4 s | 38.7 FPS |

(i.e., $[X_i, X_{i+1}^{\text{noise}}]$) to build temporal and spatial locality. Within each window, the attention from $X_i$ to $X_{i+1}^{\text{noise}}$ is excluded. We incorporate the first noisy chunk by shifting all attention and TTT masking patterns similar to Fig. 3c. The details of this hybrid architecture and more efficient trainings are in the Appendix.

# 5 Experiments

In this section, we present our experiment results on novel view synthesis (Sec. 5.1), language modeling (Sec.5.2), and autoregressive video generationo (Sec. 5.3), and an in-depth analysis (Sec. 5.4) of different design choices. Tab. 1 summarizes key factors in each experiment. When comparing with linear-cost baselines, we augmented them with the same window attention for fair comparisons.

## 5.1 Novel View Synthesis

**Datasets & metric.** We evaluate our approach on both object-level and scene-level datasets. We use Objaverse dataset [25] for object-level training, following the setup from LVSM [23] and GS-LRM [26]. After training, we perform evaluations on the Google Scanned Objects (GSO) dataset [27], at resolutions of $256 \times 256$ and $512 \times 512$. Each evaluation involves 4–48 input views and 8 novel views per object. For scene-level evaluations, we adopt the challenging DL3DV scene dataset [28], with over 11K training scenes and 140 testing scenes, each with approximately 300 views. Evaluations are at a resolution of $536 \times 960$. Performance is measured by Peak Signal-to-Noise Ratio (PSNR) at novel views, with additional metrics provided in the supplementary material.

**Model details.** Each block of model has a per-image window attention layer, a SwiGLU-MLP large-chunk TTT layer, and a feed-forward layer. The default model totals 312M parameters, including 84M fast weights ($6d^2$ per block). See Appendix for more details.

**Baselines.** For object-level evaluation, we use two baselines: a full-attention model and a Perceiver-style register-attention model [29]. The full-attention baseline replaces TTT layers with block-wise causal attention layers, enabling bidirectional interaction among input tokens and cross-attention from novel views. The Perceiver-style baseline compresses input tokens into 4096 registers, decoding novel views via cross-attention to these registers. For scene-level evaluation, we compare with LongLRM [30], a state-of-the-art model combining Mamba [12] and full attention for 3D Gaussian splat predictions, as well as pure optimization-based 3D Gaussian splatting methods. Table 2 summarizes the computational complexities of all models.

**Training details.** For object dataset, we train all models with 1.25 trillion tokens with progressive resolutions. For scene dataset, we train our model with 1.8 trillion tokens with progressively higher resolutions and more views, at a maximal sequence length of 1 million tokens. High-resolution models are trained with inner-chunk context parallelism (Sec. 3.4). See Appendix for details.

**Results.** Experimental results and analysis are presented in Figure 4.

## 5.2 Language Modeling

**Datasets & Metrics.** We train our models on the Long-Data-Collections dataset [31], using approximately 60B tokens from its total 68.8B tokens. For evaluation, we employ the per-token loss metric from [32], assessing models' ability to effectively use the full context. A monotonically

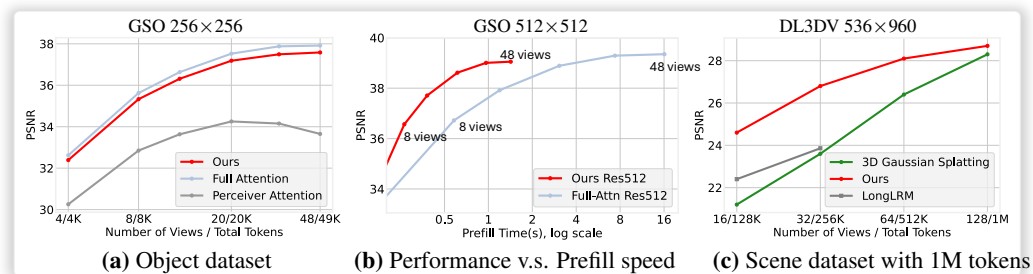

| (a) Object dataset | (b) Performance v.s. Prefill speed | (c) Scene dataset with 1M tokens |

Figure 4: **(a, b)** our method achieves quality comparable to full-attention models with significantly lower prefill latency, and it clearly outperforms perceiver-attention baselines. **(c)** On the high resolution scene dataset, our approach surpasses LongLRM, limited to 32 views, and outperforms 3D Gaussian Splatting with sparse views, remaining competitive up to 128 input views (1M total tokens).

Table 3: Comparison of baseline methods in terms of state size, training throughput (measured in tokens per second, TPS), update rules, and memory read-out mechanisms. Training throughput is evaluated using a 3B-parameter model with 32K-sequence length on A100-40GB GPUs.

|  | State size | Train TPS | Update Rule | Memory read-out |
|---|---|---|---|---|
| Transformer | – | 4.1K | – | – |
| Transformer SWA | – | 6.4K | – | – |
| *Per-token recurrence* | | | | |
| GLA SWA | $384d$ | 5.0K | $\mathbf{S}_t \leftarrow \mathbf{S}_{t-1}\mathrm{Diag}(\boldsymbol{\alpha}_t) + \mathbf{v}_t\mathbf{k}_t^\top$ | $\mathbf{o}_t = \mathbf{S}_t\mathbf{q}_t$ |
| DeltaNet SWA | $128d$ | 5.1K | $\mathbf{S}_t \leftarrow \mathbf{S}_{t-1}(\mathbf{I} - \beta_t\mathbf{k}_t\mathbf{k}_t^\top) + \beta_t\mathbf{v}_t\mathbf{k}_t^\top$ | $\mathbf{o}_t = \mathbf{S}_t\mathbf{q}_t$ |
| *Large-chunk recurrence* | | | | |
| Ours GD | $2304d$ | 5.0K | $W \leftarrow \mathrm{L2norm}(W - \sum_i^b \eta_i \nabla_W \mathcal{L}_i)$ | $\mathbf{o}_t = f_W(\mathbf{q}_t)$ |
| Ours Momentum | $2304d$ | 4.9K | $M \leftarrow \beta M + \sum_i^b \eta_i \nabla_W \mathcal{L}_i;\ W \leftarrow \mathrm{L2norm}(W - M)$ | $\mathbf{o}_t = f_W(\mathbf{q}_t)$ |
| Ours Muon | $2304d$ | 4.3K | $M \leftarrow \beta M + \sum_i^b \eta_i \nabla_W \mathcal{L}_i;\ W \leftarrow \mathrm{L2norm}(W - \mathrm{Muon}(M))$ | $\mathbf{o}_t = f_W(\mathbf{q}_t)$ |

decreasing loss indicates successful context utilization, whereas plateauing suggests limited context usage. Additionally, we report retrieval accuracy [33] at various sequence lengths.

**Model details.** We remove the window-attention layer from the original the LaCT block, integrating a sliding window-attention(SWA) layer directly into the Large-Chunk TTT layer. Following GAU [34], SWA shares Q, K, and V vectors with the fast-weight network, with additional per-channel scaling and shifting on Q and K. See supplementary for pseudocode.

**Baselines.** We compare against full attention, Gated Linear Attention (GLA) [13], DeltaNet [3, 15]. To ensure fairness, we enhance both GLA and DeltaNet with the same sliding window attention. Based on prior work [32, 35, 36] highlighting the importance of a large RoPE [37] base for long-context transformer training, we adopt a RoPE base of 1 million for training with 32K token contexts. Tab. 3 summarize the mechanism and training throughput of all methods.

**Training details.** We trained models at two scales using a 32768-token sequence length: a 760M-parameter model trained for 40B tokens with a 2048-token sliding window, and a 3B-parameter model trained for 60B tokens with a 4096-token sliding window. Further details are in the Appendix.

**Results.** Please refer to Fig. 5 for experiment results and analysis.

### 5.3 Autoregressive Video Diffusion

We fine-tune the pretrained Wan 2.1 [38] text-to-video diffusion model into an autoregressive video diffusion model. Specifically, we replace all bidirectional attention layers with our LaCT layers combined with sliding window attention. The sliding window attention uses a window size spanning two autoregressive chunks.

**Datasets.** We fine-tune the model using an internal, filtered proprietary collection of videos, each accompanied by a short text prompt generated by a visual language model.

**Training details.** Following [39, 38], we employ time-step shifting and denoising loss weighting using a logit-normal distribution. we train on 5-second videos at 16 FPS and 480×832 resolution,

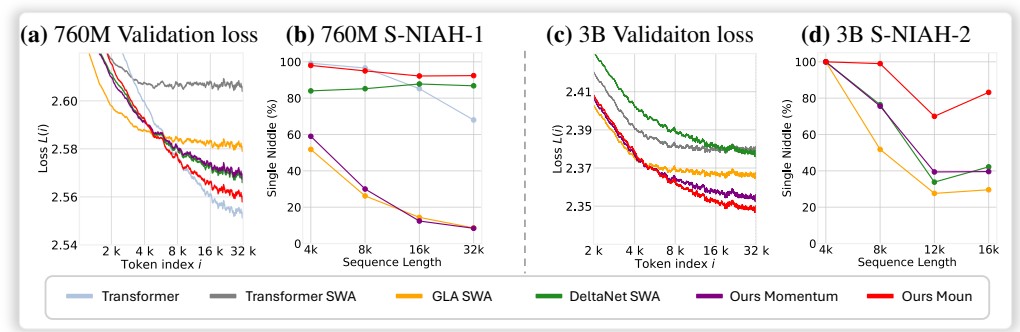

Figure 5: Language Model results. **(a, c)** Our model achieves lower per-position loss at larger token indices compared to GLA and DeltaNet at both 760M and 3B scale, indicating stronger long-context modeling capability. **(b, d)** Our model consistently outperforms GLA and DeltaNet in retrieval accuracy. Furthermore, our Muon variant consistently outperforms our Momentum variant.

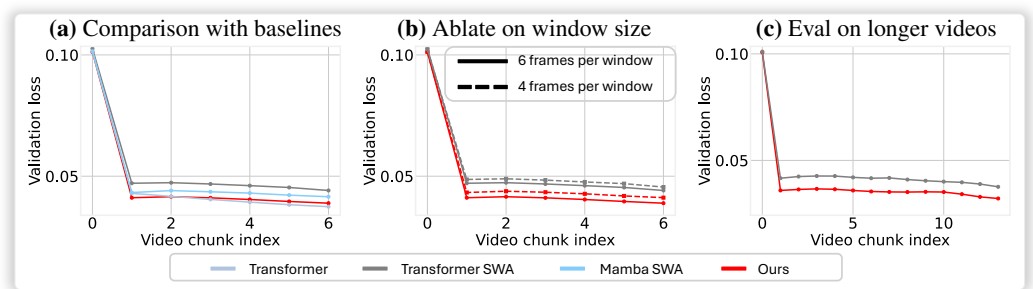

Figure 6: (a) We achieve comparable validation loss to the full-attention baseline and outperform both Mamba with sliding window and sliding window attention baselines. This improvement over SWA is consistent across different window sizes (b) and when evaluating on longer videos (c).

autoregressively denoising in 3 latent-frame chunks. Later we fine-tune the model with 8.8 second videos. See supplementary for details.

**Baselines.** We compare our method against three baselines: sliding window attention (SWA) alone, Mamba2 [20] combined with SWA (using a similar parallel combination strategy as our method), and full block-wise causal attention. Additional details are in the supplementary material.

**Evaluation.** We evaluate all models on a collection of 2,000 videos after 5,000 training iterations by computing the denoising loss at five timesteps (550, 650, 750, 850, 950). Figure 6 plots the chunk-wise denoising loss across evaluated video frames.

## 5.4 Analysis on Design Choices

In this section, we analyze several key design choices in our model, focusing on both the novel view synthesis and language modeling tasks. Specifically, we evaluate the impact of state size (Fig. 7a), test-time optimizers (Fig. 7b), linear versus nonlinear fast weights(Fig. 8a), and per-token recurrence versus chunk-wise recurrence (Fig. 8b). Please refer to section A.1 for results and conclusions.

## 6 Conclusion

We presented LaCT, a novel model architecture that integrates large-chunk test-time training for capturing long context with window attention for modeling local structure. We validated LaCT across three diverse tasks spanning different modalities—novel view synthesis, language modeling, and autoregressive video diffusion—and demonstrate its effectiveness by achieving superior or competitive performance when compared to state-of-the-art baselines. LaCT achieves high GPU efficiency even with native PyTorch implementation with dozens of lines of code and supports efficient scaling up of the state size and more flexible designs in test-time training models and optimizers. By open-sourcing the code and weights, we hope that LaCT can advocate future research explorations into more performant architectures for long-context modeling.

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

## A  Appendix

### A.1  Analysis on Design Choices

In this section, we analyze several key design choices in our model, focusing on both the novel view synthesis and language modeling tasks. Specifically, we evaluate the impact of state size (Fig. 7a), test-time optimizers (Fig. 7b), linear versus nonlinear fast weights (Fig. 8a), and per-token recurrence versus chunk-wise recurrence (Fig. 8b). Overall, we find that a large state size, advanced optimizers such as Muon, and nonlinear fast weights significantly improve our model's performance. In a controlled NVS experiment, our linear large-chunk recurrence strategy outperforms linear per-token recurrence with the same state. For language modeling, where chunk structures are not inherent, our linear large-chunk recurrence variant—while initially underperforming per-token methods like GLA and DeltaNet—surpasses them when combined with a large nonlinear state and the Muon optimizer. We refer the readers to each figure and its caption for more detailed analysis.

**Experiment details.**  The analyses in this section used the following experiment configurations:

For the NVS analysis, we utilized an object dataset, training all compared approaches for 167 billion tokens. All evaluated approaches consisted of 14 stacked blocks with a model dimension of 768.

Language modeling analyses were performed at a 760M parameter scale, training for 40 billion tokens. Both the sliding window attention (SWA) window size and LaCT chunk size were set to 2048 tokens.

For the state size scaling experiments, we keep model dimension fixed($d = 768$) and increase the intermediate multiplier of the fast weight SwiGLU MLP. For example, w an intermediate multiplier of 2 results in a hidden dimension of the fast weight SwiGLU MLP of 1536, and a total state size per block of $6d^2 \simeq 3.37$ MB.

For experiment with different test-time optimizer, our "momentum" variant follows Titans [5]. We predict a scalar momentum term $\beta_i$ from each token:

$$\beta_i = \sigma(\text{Linear}(\boldsymbol{x_i})), \tag{13}$$

where $\sigma$ is the sigmoid function. This $\beta_i$ is then averaged over all tokens in the chunk, and the average momentum is applied as follows:

$$
\begin{aligned}
g &\leftarrow \sum_i^b \eta_i \nabla_W \mathcal{L}(f_W(k_i), v_i), \\
M &\leftarrow M(\sum_i^b \beta_i/b) + g, \\
W &\leftarrow \text{weight-update}(W, M),
\end{aligned}
\tag{14}
$$

where the weight-update can be simple subtraction followed by L2 normalization normalization(as in Equation 8. ) or Muon update before subtraction(as in Equation 9.)

**Experiment details for Large-Chunk v.s. Per-token Recurrence**  . In Figure 8(b) includes controlled experiments for a fair comparison between our large-chunk recurrence and per-token recurrence strategies. In the novel view synthesis (NVS) task, "Our Linear" variant employs a linear fast weight: $f_W(q) = Wq$ and is benchmarked against a Mamba-2 baseline (a linear per-token recurrence model) with an identical state size. To accommodate the bidirectional context required by NVS over input image tokens, the Mamba-2 baseline uses two Mamba-2 layers applied in opposite directions within each model block. Both our linear variant and this bidirectional Mamba-2 have state size of $d^2$ per block. Both of these two approaches employs a per-image window attention within each model block. Under this fair comparison, our linear large-chunk recurrence achieves significantly better view synthesis performance.

For the language modeling experiments also shown in Figure 8(b), the blue line "Our Linear" variant uses the same state size ($0.25d^2$) as the GLA SWA baseline. It initially underperforms GLA SWA (blue line underperforms yellow line), likely because language data lacks the inherent chunk structures that benefit our basic linear chunk recurrence. However, when LaCT is equipped with

a larger nonlinear state ($1.5d^2$) and Muon updates, we significantly outperforms these per-token recurrence baselines.

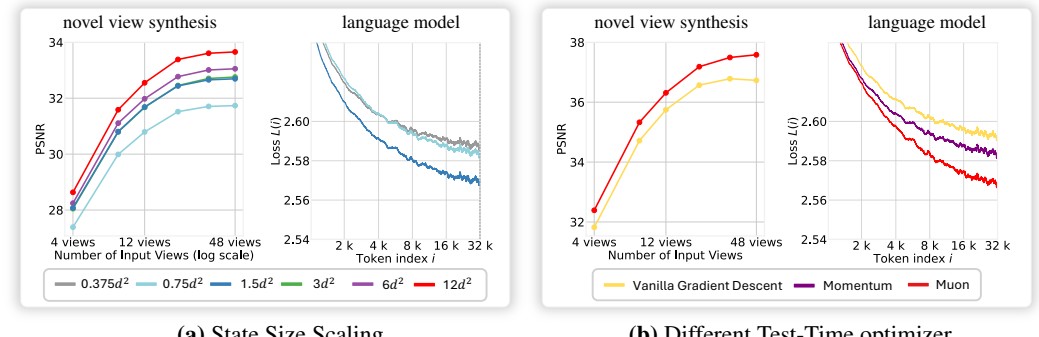

**(a)** State Size Scaling          **(b)** Different Test-Time optimizer

Figure 7: **(a)** Scaling up the state size consistently improves performance in both novel view synthesis and language modeling tasks. Note, the largest version has state size of $12d^2$ per block, totaling 40% of model weights as fast weights. **(b)** Comparison of test-time optimizers demonstrates Muon's surprising effectiveness over Vanilla Gradient Descent and Momentum.

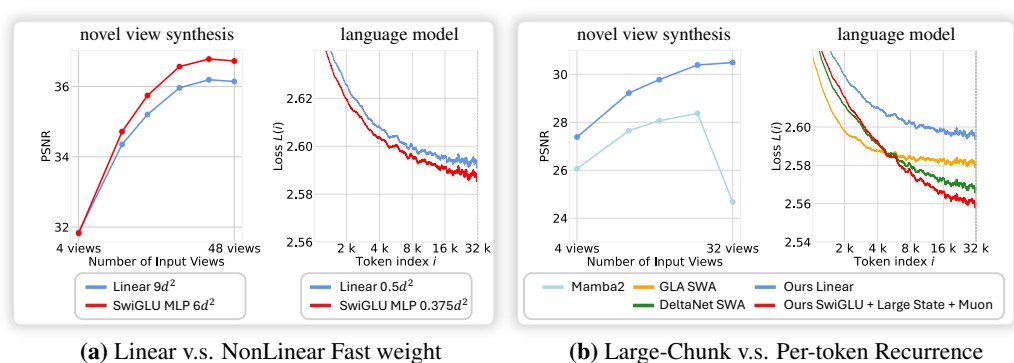

**(a)** Linear v.s. NonLinear Fast weight          **(b)** Large-Chunk v.s. Per-token Recurrence

Figure 8: **(a)** Nonlinear fast weights consistently outperform linear fast weights despite using smaller state sizes. **(b)** Our linear large-chunk recurrence approach significantly outperforms linear per-token recurrence (bidirectional Mamba2) for view synthesis tasks at the same state sizes. In language tasks, linear large-chunk recurrence of the same state size underperforms the GLA baseline, but when combined with larger nonlinear states and Muon test-time optimizer, it surpasses all per-token recurrence methods.

## A.2    Related Work

**Test-time training.** Test-Time Training (TTT) [2] is an emerging concept in sequence modeling that extends the concept of recurrent states in RNNs to online-adapted neural network components. In TTT models, a subset of weights, termed "fast weights," are updated online to store in-context information. Existing methods typically employ a self-supervised loss that encourages these fast weights to memorize key-value associations from in-context tokens, using variants of gradient descent for online adaptation.

TTT [2] [4] has opened a vast design space for new recurrent model architectures. For instance, many recent works have developed novel test-time optimizers [5, 7] and online training objectives [40]. However, current TTT approaches often suffer from low hardware utilization and limited state sizes, and consequently have not yet demonstrated their full potential. Our work primarily addresses these challenges by advocating for a new paradigm of using extremely large online minibatch (chunk) sizes for updating the fast weights. This paradigm can achieve orders-of-magnitude higher hardware utilization without relying on error-prone custom kernel implementations. Furthermore, it enables efficient scaling of nonlinear state sizes and offers the flexibility to use diverse fast weight neural networks and optimizers, thereby accelerating research progress in this area.

**Combining chunk attention with recurrence.** Several recent models combine local chunk attention with linear recurrence, such as Gated Attention Unit (GAU) [41], MEGA [42], MEGALODON [43], and InfiniAttention [44]. Among these, InfiniAttention is conceptually closest to our work, as it incorporates recurrence at the chunk level using the delta rule—interpreted as an online linear regression objective from the perspective of Test-Time Training (TTT). However, this update rule is limited in expressivity. In contrast, we employ a significantly more expressive update mechanism derived from a more general TTT framework, and demonstrate the substantial gains this brings.

Block-Recurrent Transformer [45] also explores large chunk memory updates, where memory tokens act as recurrent states that can self-attend and cross-attend with input tokens during each chunk update via attention mechanisms. The Perceiver-style register-token attention baseline used in our novel view synthesis experiments (Sec. 5.1, Table 2) is conceptually similar to the Block-Recurrent Transformer in its use of register tokens for context compression. As shown in Figure 4, our method significantly outperforms this approach in both speed and quality, with a comparable state size.

**Novel view synthesis.** Novel view synthesis (NVS) is a long-standing task at the intersection of computer vision, graphics, and computational photography, requiring algorithms to render images of a static scene from previously unobserved viewpoints. Optimization-based approaches, such as NeRF [46] and 3D Gaussian Splatting [9], have achieved significant breakthroughs. These methods optimize a set of parameterized graphics primitives (i.e., explicit or implicit representations of radiance fields) through differentiable volumetric rendering to minimize reconstruction loss on input images. After an optimization process typically lasting tens of minutes, these approaches can render novel views photorealistically, and the optimized parameters form a 3D representation of the input scene.

Recently, data-driven approaches [26, 23, 30, 47] have also shown promising results. These methods can either directly render novel views or predict 3D representations given input images. Although successful on simpler object datasets, these methods often struggle with densely sampled scenes (e.g., scenes with over 100 input images). Our experiments demonstrate that our large-chunk test-time training approach outperforms or achieves comparable performance to 3D Gaussian Splatting on challenging scene datasets with up to 128 input images with $536 \times 960$ resolution at challenging scene datasets. We hope our method will inspire further research into effectively scaling data-driven NVS methods to longer and more complex input sequences.

**Autoregressive video generation.** Current state-of-the-art video generation is dominated by bidirectional diffusion transformers operating in latent space [48, 49, 50, 38]. These models typically factorize the video distribution into a sequence of conditional distributions based on noise levels, following diffusion processes [51] or flow matching [52]. Autoregressive video generation introduces an additional temporal dimension to this factorization, where video chunks are generated iteratively, each conditioned on previously generated (and denoised) chunks.

During training, some autoregressive methods employ teacher forcing, using clean context frames and noisy subsequent frames as input [53], though this can lead to low token utilization, i.e. only a small portion of tokens get supervision. To improve token efficiency, other techniques such as progressive noise injection [54] or the use of frame-independent noises (sometimes in a diffusion-forcing style) [55, 56, 57] have been proposed. When applying our large-chunk design to autoregressive video generation, we format the input sequence with interleaved clean and noisy chunks (see Equation 12). This strategy achieves over 50% token utilization and integrates effectively with our large-chunk TTT implementation, by only changing a few lines to constrain fast-weights are only updated on clean frame chunks.

### A.3 Limitation

We conduct our experiments on three tasks. Although the tasks are diverse and cover different modalities, the effectiveness of our method would request of more tasks. For example, the novel-view synthesis task is essentially a 3D reconstruction with input pose information. The task of unposed reconstruction is more challenging and is not explored in this paper.

On the language modeling task, some key aspects are not explored due to computation limitation. These aspects include the reasoning capacity of our LaCT model and also the scalability regarding the parameter size. Previous papers showed that a main weakness of the state-based model (where

LaCT belongs to) is its reasoning ability. However, the reasoning ability is only gained with certain amount of training compute thus it is beyond our budget.

Lastly, for the autoregressive video diffusion, it is hard to find a reliable and distinguishable metric to measure the model's scalability. It is in contrast to the language modeling with perplexity (i.e., log likelihood loss) and the novel-view synthesis with PSNR. We show the validation loss in our paper and it is a common choice in evaluating the scalability of video generation. This is a general problem for the video generation evaluation and is not specific to our paper.

