# OpenReview forum: "Test-Time Training Done Right"
_NeurIPS.cc/2025/Conference — Submitted to NeurIPS 2025_

### Official Review · Reviewer_Zdxy · 2025-06-13

**Clarity:** 3
**Significance:** 4
**Originality:** 4
**Rating:** 5
**Confidence:** 3

**Summary:**

Test time training has the goal of increasing model efficiency and performance by adapting a subset of network parameters during inference. This paper proposes a new approach to this problem by introducing algorithmic and architectural innovations that allow for the use of large chunk sizes.

**Questions:**

- The maximum number of TTT parameters tested is about 40% of the total number of model parameters. Can the authors provide experiments to show the effect of further increases in the number of parameters or discuss the potential of this change.

**Ethical Concerns:**

["NO or VERY MINOR ethics concerns only"]

**Final Justification:**

I recommend acceptance because of introducing novelties in a new domain and a new presentation of modern TTT.

**Limitations:**

Suggestions to increase clarity:
-Fig. 3 is not clear. I agree that the decoupling between the update and apply functions allows for interesting combinations but what do the blue squares indicate and how are they linked to the architectures shown above?
- Fig. 3: the arrow from update to apply should be labeled f(w) to avoid confusion.
- In the video diffusion experiment, the frozen weights are used for denoising. Please clarify the role played by the TTT wights.
- The L2 norm equation for weight update should be introduced earlier in 2.1
- Mention the GPU resources used

**Quality:**

4

**Strengths And Weaknesses:**

Strengths:
- The proposed approach is general and can be utilized and adapted to a variety of different contexts and applications.
- It can be easily integrated with current deep learning libraries.
- The performance was demonstrated in multiple challenging tasks.

Weaknesses:
- Clarity of presentation can be improved as the paper currently requires a very good familiarity with TTT.
- The maximum number of TTT parameters tested is about 40% of the total number of model parameters. The authors should provide experiments to show the effect of further increases in the number of parameters or discuss the potential of this change.

---

> ### Author Rebuttal · Authors · 2025-07-31
>
> We greatly appreciate the reviewer's thoughtful feedback. Below, we clarify and address each point raised:
>
> **Reviewer comment:**
> > Clarity of presentation can be improved as the paper currently requires a very good familiarity with TTT.
>
> **Response:**
> 1. Move the related work section on Test-Time Training (TTT) from the appendix into the main paper, aiding readers less familiar with TTT.
>
> 2. Include clear cross-references and direct links between the main paper and appendix sections, as also recommended by reviewer JVw8.
>
> 3. Move critical ablations (e.g., Figures 7 and 8) from the appendix to the main body, while relocating extensive technical details into supplementary material to streamline readability.
>
> **Reviewer comment:**
> > The maximum number of TTT parameters tested is about 40% of the total number of model parameters. The authors should provide experiments to show the effect of further increases in the number of parameters or discuss the potential of this change.
>
> **Response:**
> We thank the reviewer for highlighting this promising direction. Our current experiments already substantially expand TTT parameter scales beyond existing methods:
>
> For example, the DeltaNet baseline (760M parameters) uses state sizes about 0.7% of total parameters per block, whereas our largest experiments reach up to 40%.
>
> Figure 7(a) clearly demonstrates performance gains with increased state sizes, where novel view synthesis improves consistently from 4.4% to 40% TTT parameters, and language modeling tasks improve from 3% to 11%.
>
> We agree that exploring even larger state sizes remains exciting and significant for future research.
>
> **Reviewer comment:**
> > Suggestions to increase clarity: -Fig. 3 is not clear. I agree that the decoupling between the update and apply functions allows for interesting combinations but what do the blue squares indicate and how are they linked to the architectures shown above?
>
> **Response:**
> We appreciate the reviewer's suggestion. Yes, it's a little ambiguous.
>
> The small blue squares in Fig. 3 represent valid attention mask positions, where rows correspond to query tokens and columns to key tokens. A blue square at [row, column] indicates the ```column```-th key token is visible to the ```row```-th query token. We will update the figure caption accordingly for clarity.
>
> We will adjust the caption of the figure to explain the meaning of the blue square.
>
> **Reviewer comment:**
> > In the video diffusion experiment, the frozen weights are used for denoising. Please clarify the role played by the TTT weights.
>
> **Response:**
> The TTT weights specifically store temporary memory from previously denoised (clean) video frames. During inference, noisy video chunks query these frozen TTT weights to retrieve contextual information from earlier frames. After the full denoising process for each chunk of video frames, TTT weights update based on the most recent clean chunk, effectively functioning as block-wise causal memory.
>
>
> **Reviewer comment:**
> > The L2 norm equation for weight update should be introduced earlier in 2.1
>
> **Response:**
> We believe the current positioning of the L2 norm equation in Section 3.2 remains appropriate. Section 2.1 references original equations from the previous TTT literature, whereas Section 3 explicitly introduces our LaCT-specific design involving weight normalization.
>
> **Reviewer comment:**
> > Mention the GPU resources used
>
> **Response:**
> We mentioned the GPU resources for major experiments in each modality in Section E of the Supplementary material.  For reference:
>
> For novel view synthesis:
> The training is completed with 64 A100 GPUs. The pre-training takes around 8 days, and each fine-tuning stage is about 12hours (thus 2 days in total).
>
> For language model experiments
> - 760M parameters scale experiment:  Each experiment ran on 32 A100-40GB SXM GPUs for approximately 20 hours.
> -  3B parameters scale experiments: . Each experiment ran on 64 A100-40GB SXM GPUs for approximately 50-60 hours.
>
> For autoregressive video generation experiments:
> - 1.3 B parameter scale experiments: 64 A100 80GB SXM GPUs for around 35 hours
> - 14B parameter scale experiments:  the 5-seconds video training takes approximately 110 hours on 64 A100 80GB SXM, then fintuned for 8.8 secs video for an additionally 13 hours on 64 H100 80GB SXM.

---

> > ### Comment · Reviewer_Zdxy · 2025-08-03
> >
> > Thanks for your feedback.

---

### Official Review · Reviewer_JVw8 · 2025-06-26

**Clarity:** 2
**Significance:** 1
**Originality:** 1
**Rating:** 2
**Confidence:** 3

**Summary:**

This paper proposes to scale up the existing "test-time training" (TTT) paradigm from [1] by extending this methodology with large batch sizes across many inputs in parallel. The central idea is to combat the poor observed FLOP utilization of TTT with small batches without custom kernels. The resulting pipeline, named LaCT (large chunk TTT), is tested primarily on novel view synthesis and also on other different applications with transformers, as language and video modeling.,

[1] https://arxiv.org/abs/2310.13807

**Questions:**

1. In contrast to the paper's initial claim, the original TTT implementation from [1] already seems to use JAX compilation for kernel fusion and a few updates (up to 4) with a very large batch size in their pixel imagenet experiment. In fact, they show almost equivalent cost of their TTT layer with standard linear attention. Am I missing something?

[1] https://arxiv.org/abs/2310.13807

Overall, while I believe the paper shows some potentially interesting results applying TTT on novel view synthesis, it seems to lack several components to support its very specific contribution (e.g., ablations and analysis of the novel design decisions), and seems to have numerous areas of clear improvement as outlined above. Thus, I do not believe the current submission is ready for acceptance.

**Ethical Concerns:**

["NO or VERY MINOR ethics concerns only"]

**Final Justification:**

I still find that most of the criticism I raised in my review remains valid.

"We respectfully clarify that our contribution extends significantly beyond minor hyperparameter modifications. Previous TTT methods (e.g., TTT-linear, TTT-MLP) have severe practical limitations. For instance, the original open-source TTT-linear implementation requires 32 A100 GPUs for 95 days for our smallest language modeling experiment (760M parameters, 40B tokens), whereas LaCT completes this in ~20 hours on identical hardware."

I find this statement once again potentially misleading. The Pytorch TTT implementation shared by the original authors has not been optimized, has not been used for training, and is only meant for reference purposes. In fact, at the top of the README, they even state:

"The original Pytorch implementation of TTT was never optimized for inference [...] We do not recommend training with this codebase, because it is written in pure PyTorch without any systems optimization, so training will be slow, especially when the per-device batch size is small.

For training code, or to replicate results from our paper, please view our JAX codebase. For inference kernels, or to replicate speed benchmarks from our paper, please view our kernel implementations."

Beyond the hyperparameters, if the technical innovation is the "faster" Pytorch re-implementation (which was not shared at the time of submission), I do not think this matches the paper's claimed contributions.

As I mentioned in my review, I think some applications of the TTT framework, such as novel view synthesis, are interesting and should be expanded further. However, both the original paper and the rebuttal claims appear quite significantly inconsistent with the paper's actual contribution. Thus, I cannot recommend the paper for acceptance.

**Limitations:**

Many of the above-mentioned limitations, such as the lack of proper ablations and analysis, are never explicitly discussed. Instead, the authors mention aspects that do not even seem as limitations in comparison e.g.: lines 566-567 "We conduct our experiments on three tasks. Although the tasks are diverse and cover different  modalities, the effectiveness of our method would request of more tasks."

Moreover, the authors never discuss societal implications. Their justification seems weak "This is a general method paper and it does not have specific concerns regarding 807 societal impacts comparing with other papers."

**Paper Formatting Concerns:**

The paper has some noticeable formatting violations, including all section headings using the wrong capitalization.

**Quality:**

2

**Strengths And Weaknesses:**

Strengths.

1. The paper targets three relevant empirical domains to validate their scaled-up version of TTT. The most significant of the results appears to be for the view synthesis application, which shows TTT can achieve only slightly inferior performance to full-attention transformers.
2. The paper introduces the prior work of the TTT paradigm and its efficiency challenges when considering 'naive' Python implementations, as well as the traditional need for specialized kernels, both succinctly and clearly (Section 2).
3. Some of the figures are explanations (e.g., ordering in of apply and shift operations to simulate a block causal mask in Figure 3) are intuitive and would again provide a nice explanation for some implementation decisions needed for applying TTT to different domains,

Weaknesses and areas for improvement.

1. The core contribution appears as minor hyperparameter changes, such as substituting the 2-layer MLP in (2) with a SwiGLU 2-layer MLP, and the slight change in the fast weight loss, removing a |f_W|^2 term from the L2.
2. Claims in the checklist appear to be poorly justified or even completely false. For instance, the authors answered [YES] to the questions regarding code sharing and reproducibility. Yet, they did not provide any code with the submission, with this claim being "justified" by a promise of sharing code conditioned on acceptance.
3. Related to the point above, as I believe that the main paper's advancement is an efficient implementation, rather than a new method, algorithm, or idea, its contribution largely hinges on the author's promise of sharing the code upon acceptance.


4. The central claim that the proposed method improves "improves hardware utilization by orders of magnitude" in the Appendix (e.g., lines 15-16) seems to be entirely dependent on the specific hardware (an A100 GPU in this case). This claim should be properly contextualized to avoid potentially misleading a non-familiar audience. Other claims in the paper, such as "Language Modeling. Our model achieves competitive performance compared to SoTA methods such as DeltaNet," (lines 62-63) also seem potentially misleading. DeltaNet is a model that has been partially validated mostly on very small-scale language modeling and synthetic tasks like MAD [2].

5. Many critical details are deferred to the Appendix. However, references to these details are never specific. E.g., lines 240, 252, ... "See Appendix for details." At least the authors should have provided Section numbers to facilitate parsing the text. Moreover, I could not even find many of these details anywhere e.g., lines 51-53 "LaCT significantly enhances parallelism, leading to substantially improved GPU utilization with just a few dozen lines of pure PyTorch code (see the Appendix)." There seems to be no lines of Python code in the appendix.

6. While most of the contributions appear to be changes in architectural and other small hyperparameter choices (e.g., SwiGLU vs 2-layer MLP, new loss function etc.) There is a lack of proper validation with, for instance, target ablations and other kinds of small-scale analysis.

[1] https://arxiv.org/abs/2310.13807
[2] https://arxiv.org/abs/2403.17844

---

> ### Author Rebuttal · Authors · 2025-07-31
>
> We appreciate the detailed review and provide clarifications and responses below.
>
>
> ### Reviewer Concern 1:
> > "The core contribution appears as minor hyperparameter changes..."
>
> **Response:**
> We respectfully clarify that our contribution extends significantly beyond minor hyperparameter modifications. Previous TTT methods (e.g., TTT-linear, TTT-MLP) have severe practical limitations. For instance, the original open-source TTT-linear implementation requires 32 A100 GPUs for 95 days for our smallest language modeling experiment (760M parameters, 40B tokens), whereas LaCT completes this in ~20 hours on identical hardware.
>
> The primary contribution is our proposed LaCT framework, which enables efficient experimentation across diverse design spaces—including test-time optimizers, fast weight architectures, loss functions, and modalities—without cumbersome custom kernels or extensive modifications. Unlike approaches such as GLA or DeltaNet, which heavily depend on advanced Triton kernels requiring specialized expertise(which only a small group of researchers can write high-quality triton code), LaCT's pure PyTorch implementation significantly democratizes access and accelerates research in efficient nonlinear recurrent models.
>
>
>
> ### Reviewer Concern 2 & 3:
>
> >"Claims about code sharing and reproducibility seem poorly justified or false."
>
> **Response:**
> **We will publicly release our code regardless of acceptance**.
> Due to the ban of anonymous github links in rebuttal, we can paste some key part of code in chat if necessary.
>
> We have internally prepared an open-source implementations covering:
>
> 1. Novel view synthesis model, training and inference code, with checkpoints for both object and scene dataset.
>
> 2. Language model code with huggingface transformer style (which is compitable with lot's of open-source trainer).
>
> 3. AutoRegressive Video diffusion model, training and inference code.
> 4. A minimal implementation of model code rapid prototyping and experimentation.
>
>
> ### Reviewer Concern 4(a):
>
> >"The central claim that the proposed method improves "improves hardware utilization by orders of magnitude" in the Appendix (e.g., lines 15-16) seems to be entirely dependent on the specific hardware (an A100 GPU in this case). This claim should be properly contextualized to avoid potentially misleading a non-familiar audience."
>
> **Response:**
> We tested throughput on both A100 and H100 GPUs (see Figure 1(a)). We will contextualize hardware specifics clearly in the revision.
>
> Importantly, our PyTorch implementation is GPU-agnostic, requiring no device-specific customization, suggesting broad applicability across modern GPUs.
>
>
>
> ### Reviewer Concern 4(b):
>
> >"Other claims in the paper, such as "Language Modeling. Our model achieves competitive performance compared to SoTA methods such as DeltaNet," (lines 62-63) also seem potentially misleading."
>
> **Response:**
> We will adjust the wording from "competitive performance compared to SoTA methods" to "competitive performance compared to current popular methods like DeltaNet."
>
>
>
> ### Reviewer Concern 5(a):
>
> >"Critical details deferred to Appendix without specific references."
>
> **Response:**
> We appreciate this feedback and will add explicit section references and hyperlinks in the revised manuscript to clearly guide readers to relevant details.
>
>
>
> ### Reviewer Concern 5(b):
>
> >"There seems to be no lines of Python code in the appendix."
>
> **Response:**
> We already included Python-style pseudocode in **Section C (pp. 15-16)** of the submitted supplementary material. (Please unzip the supplementary material to see the pdf named “supplementary_materials_TTTDR_neurips.pdf”.
>
>
>
> ### Reviewer Concern 6:
>
> > "Lack of proper ablations and small-scale analyses."
>
> **Response: **
>
> Our paper already provides several targeted ablation studies isolating key design choices on both novel view synthesis and language modeling tasks:
>
> - Figure 7(a): Larger state size consistent boost performance
>
> - Figure 7(b): Advanced test-time training optimizers like Muon and Momentum boost performance.
>
> - Figure 8(a): Nonlinear fast-weight functions outperforms linear fast weight
>
> - Figure 8(b): Evaluation of chunk recurrence versus per-token recurrence on different data structure.
>
>
>
> ### Reviewer Question:
>
> > "The original TTT implementation [1] uses JAX compilation for large batch sizes and reports comparable FLOPs with linear attention. What am I missing?"
>
> **Response:**
> The reviewer references Table 1 from [1], where TTT-linear FLOPs are 1.1× linear attention, and TTT-MLP is 1.5× linear attention. However, **FLOPs alone poorly reflect real-world GPU runtime performance**. Real runtime speed heavily depends on arithmetic intensity and parallelism, not just FLOPs.
>
> For example, in our experiments, although TTT-linear theoretically has fewer FLOPs than Transformers, its runtime is around 90x longer  (90 days on 32 A100 GPUs) versus Transformers (within 1 day, same hardware). LaCT resolves this severe practical inefficiency, dramatically reducing the test-time training runtime to ~20 hours and enabling feasible, extensive experimentation.
>
> Moreover, while JAX compilation on TPUs can fuse kernels and reduce memory overhead, achieving orders-of-magnitude speedup is uncommon. Additionally, most researchers, especially within the open-source community, predominantly use PyTorch and GPUs, environments that LaCT directly supports.

---

> ### Comment · Reviewer_JVw8 · 2025-08-04
> **Post rebuttal**
>
> I would like to thank the authors for their rebuttal. Unfortunately, I still find that most of the criticism I raised in my review remains valid.
>
> "We respectfully clarify that our contribution extends significantly beyond minor hyperparameter modifications. Previous TTT methods (e.g., TTT-linear, TTT-MLP) have severe practical limitations. For instance, the original open-source TTT-linear implementation requires 32 A100 GPUs for 95 days for our smallest language modeling experiment (760M parameters, 40B tokens), whereas LaCT completes this in ~20 hours on identical hardware."
>
> I find this statement once again potentially misleading. Please correct me if I am looking at the wrong GitHub codebase, but the Pytorch TTT implementation shared by the original authors has not been optimized, has not been used for training, and is only meant for reference purposes. In fact, at the top of the README, they even state:
>
> "The original Pytorch implementation of TTT was never optimized for inference [...] We do not recommend training with this codebase, because it is written in pure PyTorch without any systems optimization, so training will be slow, especially when the per-device batch size is small.
>
> For training code, or to replicate results from our paper, please view our JAX codebase. For inference kernels, or to replicate speed benchmarks from our paper, please view our kernel implementations."
>
> Beyond the hyperparameters, if the technical innovation is the "faster" Pytorch re-implementation (which was not shared at the time of submission), I do not think this matches the paper's claimed contributions.
>
> As I mentioned in my review, I think some applications of the TTT framework, such as novel view synthesis, are interesting and should be expanded further. However, both the original paper and the rebuttal claims appear quite significantly inconsistent with the paper's actual contribution. Thus, I cannot recommend the paper for acceptance.

---

> > ### Author Response · Authors · 2025-08-05
> > **Round-2 reply**
> >
> > We appreciate the reviewer’s prompt reply and address the remaining concerns below:
> >
> > **Concern 1: Original TTT Implementation Efficiency on Jax**
> >
> > The publicly available JAX implementation encounters out-of-memory issues when applied to language modeling at the scale of our smallest experiment (760M parameters, 32K context length, 40B tokens). When training at shorter sequence length,(e.g., 8K tokens for the same total data), it takes 32 A100-80GB **8.9 days** for TTT-Linear and **22.5 days** for TTT-MLP. In contrast, our proposed LaCT and other baselines achieves equivalent training in ~20 hours on identical hardware, offering a near 10x practical speedup. It’s nearly impossible to scale-up and ablate on a baseline that runs nearly 10x slower!
> >
> > **Concern 2: Claims Regarding Code Release**
> >
> > The reviewer expressed concerns regarding our checklist responses about reproducibility and code sharing, suggesting these claims were unjustified or potentially false due to the absence of provided code in the submission (quotes: "Claims in the checklist appear to be poorly justified or even completely false").
> >
> >
> >
> > To clarify, I copy the checklist claims about reproducibility and core sharing here:
> >
> > >Claim-4.
> >  **Question:** Does the paper fully disclose all the information needed to reproduce the main experimental results of the paper to the extent that it affects the main claims and/or conclusions of the paper (regardless of whether the code and data are provided or not)?  **Answer:** [Yes]. **Justification:** Yes, we provided. We will also release the code and model checkpoints for some of the tasks.
> >
> > >Claim-5. **Question:** Does the paper provide open access to the data and code, with sufficient instructions to faithfully reproduce the main experimental results, as described in supplemental material?  **Answer:** [Yes]. **Justification:** We present pseudo code in supp and will release the code once acceptance.
> >
> >  These checklist criteria **did not** explicitly mandate code submission during the review stage. Therefore, we respectfully ask the reviewer to specify the exact points where our claims regarding code-sharing and reproducibility were poorly justified or incorrect.
> >
> > Recognizing the importance of transparency and code availability:
> >
> > - We guarantee open-source release of all code regardless of acceptance decision, in any of our public version.
> > - We are prepared to directly share substantial code segments during the discussion phase within the character limits if requested.

---

> > > ### Comment · Reviewer_JVw8 · 2025-08-07
> > > **Post reply**
> > >
> > > As the authors appear to focus on their code, I will try to be clearer:
> > >
> > > If the authors found some undocumented issue in the Jax codebase, which does not match the TTT paper's efficiency claims without minor tweaks, I would suggest they raise it as such. I would suggest avoiding claiming that speeding up TTT N times, thanks to a new Pytorch implementation to be the paper's main contribution. I believe this is not only partially out-of-scope for a NeurIPS paper's main contribution, but it would also require at least providing reviewers with the code at the time of submission.
> > >
> > > Once again, I believe some of the applications of the TTT framework, such as novel view synthesis, are interesting. I believe these should be expanded further to make the paper's contribution, beyond parameter changes, more concrete.
> > >
> > > Minor: Regarding the checklist, I want to avoid copying and pasting the full guidelines right below the text they quoted. For instance:
> > >
> > > "At submission time, to preserve anonymity, the authors should release anonymized versions (if applicable)"

---

> > > > ### Author Response · Authors · 2025-08-08
> > > > **Round-3 reply**
> > > >
> > > > We thank the reviewer for the prompt reply and engagement. Here is our new response.
> > > > ---
> > > > Regarding the  claims about  speeding up TTT.
> > > >
> > > > The original TTT paper [1] did not report GPU training speed for language models—only TPU training with JAX and GPU inference using specialized inference kernels.
> > > >
> > > > We therefore measured training speed using the official PyTorch (reference, not recommended for training) and JAX code on A100 GPUs. For a 760M-parameter LM with 32K context and 40B tokens, TTT-Linear in JAX required ~8.9 days on 32×A100-80GB GPUs, whereas LaCT completed the same task in ~20 hours on identical hardware.
> > > >
> > > > We also compared the training speed of its customized triton and TK kernels in Figure-1 of our submission. Ours simple pytorch implementation also achieves much higher FLOPS utilization.    (note the TTT training kernel does not support the per-token causal mask required for language models, so these kernels cannot be used for training of the original LM setup in [1].)
> > > >
> > > > From the theotical perspective,  original TTT computes test-time gradients every 16–64 tokens, yielding arithmetic intensity lower than 16 or 64, far below modern GPU thresholds (155 bytes/FLOP on A100, 292 on H100), making it inherently memory-bound regardless of whether it is implemented in PyTorch, JAX, Triton, or even CUDA code.
> > > >
> > > > By contrast, LaCT is explicitly designed for higher arithmetic intensity and hardware friendliness. Our method requires only a few dozen lines of simple PyTorch code (pseudo-code in Supplementary, pp. 15–16) to achieve significantly better FLOPS utilization.
> > > >
> > > > Thus, we believe we can argue that our LaCT by design is more hardware friendly and can achiever faster training on modern GPUs.
> > > >
> > > > [1] Learning to (Learn at Test Time): RNNs with Expressive Hidden States. Yu et. al.
> > > >
> > > > ---
> > > > The reviewer criticized us for not being able to provide code at the time of submission.
> > > >
> > > > We understand the importance of open-sourcing code and appreciate your high standards on code release—we believe this is great for the community.
> > > >
> > > > Compared to the triton and tk kernel of the original test-time training paper, our pytorch code are much simpler to implement, and we have a pytorch-stype pesudocode at the time of submission at Page 15-16 of the supplementary material.  The simplicity of our design is itself part of the contribution—it avoids cumbersome kernels, lowering the barrier to applying TTT to new domains, or exploring different optimizers or architecture in the TTT design space.
> > > >
> > > > We will also append our major PyTorch code in the following reply. (We apologize for the forthcoming long replies.)

---

> > > > > ### Author Response · Authors · 2025-08-08
> > > > > **Pytorch Code  Part-1:  apply and update operation of lact used in llm and ar video diffusion**
> > > > >
> > > > > Minimal implementation of the apply and update operation for the block causal LaCT with SwiGLU fast weight function.
> > > > > ```python
> > > > > @torch.compile()
> > > > > def zeropower_via_newtonschulz5(G):
> > > > >     """
> > > > >     The code is most copied from xxx [url link removed due to rebuttal constraints].
> > > > >     Args:
> > > > >         G: [b, d, d']
> > > > >     Returns:
> > > > >         X: [b, d, d']
> > > > >     """
> > > > >     assert len(G.shape) == 3
> > > > >     X = G.bfloat16()
> > > > >     if G.size(1) > G.size(2):
> > > > >         X = X.transpose(1, 2)
> > > > >     X = X / (X.norm(dim=(1, 2), keepdim=True) + 1e-7)
> > > > >     for a, b, c in [
> > > > >         (4.0848, -6.8946, 2.9270),
> > > > >         (3.9505, -6.3029, 2.6377),
> > > > >         (3.7418, -5.5913, 2.3037),
> > > > >         (2.8769, -3.1427, 1.2046),
> > > > >         (2.8366, -3.0525, 1.2012),
> > > > >     ]:
> > > > >         A = X @ X.transpose(1, 2)
> > > > >         B = b * A + c * A @ A
> > > > >         X = a * X + B @ X
> > > > >
> > > > >     if G.size(1) > G.size(2):
> > > > >         X = X.transpose(1, 2)
> > > > >     return X
> > > > >
> > > > >
> > > > > @torch.compile
> > > > > def block_causal_lact_swiglu(
> > > > >     w0: torch.Tensor,  # [b, dh, dk]
> > > > >     w1: torch.Tensor,  # [b, dv, dh]
> > > > >     w2: torch.Tensor,  # [b, dh, dk]
> > > > >     q: torch.Tensor,  # [b, l, dk]
> > > > >     k: torch.Tensor,  # [b, l, dk]
> > > > >     v: torch.Tensor,  # [b, l, dv]
> > > > >     lr0: torch.Tensor,  # [b, l, 1]
> > > > >     lr1: torch.Tensor,  # [b, l, 1]
> > > > >     lr2: torch.Tensor,  # [b, l, 1]
> > > > >     momentum: torch.Tensor = None,  # [b, s, 1] # none means no momentum
> > > > >     use_muon: bool = True,
> > > > >     chunk_size: int = 2048,
> > > > > ):
> > > > >     """
> > > > >     Block causal LaCT with SwiGLU fast weight function.
> > > > >         Apply then Update => Shifted Block Causal LaCT
> > > > >     w0, w1, w2 are the fast weights. f(x) =  w1 @ (silu(w0 @ x) * (w2 @ x))
> > > > >     """
> > > > >     w0_norm = w0.norm(dim=2, keepdim=True)
> > > > >     w1_norm = w1.norm(dim=2, keepdim=True)
> > > > >     w2_norm = w2.norm(dim=2, keepdim=True)
> > > > >
> > > > >     if momentum is not None:
> > > > >         dw1_momentum = torch.zeros_like(w1)
> > > > >         dw0_momentum = torch.zeros_like(w0)
> > > > >         dw2_momentum = torch.zeros_like(w2)
> > > > >
> > > > >     q = q.transpose(1, 2)  # [b, dk, l]
> > > > >     v = v.transpose(1, 2)
> > > > >     output = torch.zeros_like(v)
> > > > >     e_index = 0
> > > > >     seq_len = k.shape[1]
> > > > >     for i in range(0, seq_len - chunk_size, chunk_size):
> > > > >         s_index = i
> > > > >         e_index = s_index + chunk_size
> > > > >
> > > > >         # [b, l, dk]
> > > > >         ki = k[:, s_index:e_index, :]  # bf16
> > > > >         # [b, dv, l]
> > > > >         vi = v[:, :, s_index:e_index]  # bf16
> > > > >         # [b, dh, l]
> > > > >         qi = q[:, :, s_index:e_index]
> > > > >         # [b, l, d/1] fp32
> > > > >         lr1i = lr1[:, s_index:e_index, :]  # [b, l, d/1] fp32
> > > > >         lr2i = lr2[:, s_index:e_index, :]
> > > > >         lr0i = lr0[:, s_index:e_index, :]
> > > > >
> > > > >         # [b, dh, dk] @ [b, dk, l] -> [b, dh, l]
> > > > >         h = torch.bmm(w2, qi)
> > > > >         gate = F.silu(torch.bmm(w0, qi), inplace=True)
> > > > >         # [b, dv, dh] @ [b, dh, l] -> [b, dv, l] -> [b, l, dv]
> > > > >         output[:, :, s_index:e_index] = torch.bmm(w1, gate * h)
> > > > >
> > > > >         ########## compute the gradient and update the fast weights
> > > > >         # [b, dh, dk] @ [b, dk, l] -> [b, dh, l]
> > > > >         gate_before_act = torch.bmm(w0, ki.transpose(1, 2))
> > > > >         hidden_before_mul = torch.bmm(w2, ki.transpose(1, 2))
> > > > >         hidden = F.silu(gate_before_act, inplace=False) * hidden_before_mul
> > > > >
> > > > >         # [b, dh, dv] @ [b, dv, l] -> [b, dh, l]
> > > > >         dhidden = torch.bmm(w1.transpose(1, 2), vi)
> > > > >
> > > > >         dhidden_before_mul = dhidden * F.silu(gate_before_act, inplace=False)
> > > > >
> > > > >         dgate = dhidden * hidden_before_mul
> > > > >         dgate_before_act = silu_backprop(dgate, gate_before_act)
> > > > >
> > > > >         # [b, dv, l] @ [b, l, dh] -> [b, dv, dh]
> > > > >         # it's better to cast the mat to bf16 before bmm.
> > > > >         dw1 = torch.bmm(vi, (hidden.transpose(1, 2) * lr1i).type_as(vi))  # [b, d, d]
> > > > >         # [b, dh, l] @ [b, l, dk] -> [b, dh, dk]
> > > > >         dw0 = torch.bmm(dgate_before_act, (ki * lr0i).type_as(dgate_before_act))
> > > > >         dw2 = torch.bmm(dhidden_before_mul, (ki * lr2i).type_as(dhidden_before_mul))
> > > > >
> > > > >         if momentum is not None:
> > > > >             m_i = momentum[:, s_index:e_index, :]
> > > > >
> > > > >             m_i = m_i.mean(dim=1, keepdim=True)
> > > > >
> > > > >             dw0 = dw0 + dw0_momentum * m_i
> > > > >             dw1 = dw1 + dw1_momentum * m_i
> > > > >             dw2 = dw2 + dw2_momentum * m_i
> > > > >             dw0_momentum = dw0
> > > > >             dw1_momentum = dw1
> > > > >             dw2_momentum = dw2
> > > > >
> > > > >         if use_muon:
> > > > >             dw1 = zeropower_via_newtonschulz5(dw1)
> > > > >             dw0 = zeropower_via_newtonschulz5(dw0)
> > > > >             dw2 = zeropower_via_newtonschulz5(dw2)
> > > > >
> > > > >         w0 = w0 + dw0
> > > > >         w1 = w1 + dw1
> > > > >         w2 = w2 + dw2
> > > > >         w0 = w0 / (w0.norm(dim=2, keepdim=True) + 1e-5) * w0_norm
> > > > >         w1 = w1 / (w1.norm(dim=2, keepdim=True) + 1e-5) * w1_norm
> > > > >         w2 = w2 / (w2.norm(dim=2, keepdim=True) + 1e-5) * w2_norm
> > > > >
> > > > >     s_index = e_index  # update_length - mini_batch_size
> > > > >     e_index = seq_len
> > > > >     qi = q[:, :, s_index:e_index]
> > > > >     # get the final output
> > > > >     # [b, dh, dk] @ [b, dk, l] -> [b, dh, l]
> > > > >     h = torch.bmm(w2, qi)
> > > > >     gate = F.silu(torch.bmm(w0, qi), inplace=True)
> > > > >     # [b, dv, dh] @ [b, dh, l] -> [b, dv, l]
> > > > >     output[:, :, s_index:e_index] = torch.bmm(w1, gate * h)
> > > > >     # [b, l, dv]
> > > > >     return output.transpose(1, 2)
> > > > >
> > > > > ```

---

> > > > > > ### Author Response · Authors · 2025-08-08
> > > > > > **Pytorch Code Part-2: apply and update operation of bidirectional lact used in view synthesis experiment**
> > > > > >
> > > > > > The Update and Apply operations of the Bidirectional LaCT used in the novel view synthesis experiment. The `zeropower_via_newtonschulz5`  function is included in part-1.
> > > > > > Since I have more space for this reply, I have added more detailed comments in the code.
> > > > > >
> > > > > > ```python
> > > > > > @torch.compile
> > > > > > def bidirectional_lact_swiglu(
> > > > > >     w0: torch.Tensor,  # [b, dh, dk]
> > > > > >     w1: torch.Tensor,  # [b, dv, dh]
> > > > > >     w2: torch.Tensor,  # [b, dh, dk]
> > > > > >     q: torch.Tensor,  # [b, l, dk]
> > > > > >     k: torch.Tensor,  # [b, l, dk]
> > > > > >     v: torch.Tensor,  # [b, l, dv]
> > > > > >     lr0: torch.Tensor,  # [b, l, 1]
> > > > > >     lr1: torch.Tensor,  # [b, l, 1]
> > > > > >     lr2: torch.Tensor,  # [b, l, 1]
> > > > > >     use_muon: bool = True,
> > > > > > ) -> torch.Tensor:
> > > > > >     """
> > > > > >     Bidirectional LaCT with SwiGLU fast weight function.
> > > > > >     w0, w1, w2 are the fast weights. f(x) =  w1 @ (silu(w0 @ x) * (w2 @ x))
> > > > > >
> > > > > >     About precision:
> > > > > >         w0, w1, w2 are mostly likely fp32.
> > > > > >         q, k, v are fp16.
> > > > > >         lr0, lr1, lr2 are fp32.
> > > > > >         The forward, backward produce bf16 gradients, updated fast weights are fp32.
> > > > > >         The final output are bf16.
> > > > > >
> > > > > >
> > > > > >     FLOPS:
> > > > > >         (assume dk=dv denoted as D, hidden dimension of swiglu-mlp is H, ignore muon)
> > > > > >         Forward pass with key: 4 * D * H * L * B
> > > > > >         Backward pass: 8 * D * H * L * B
> > > > > >         Forward with Query: 6 * D * H * L * B
> > > > > >         Total: 18 * D * H * L * B
> > > > > >     Outputs:
> > > > > >         o: [b, l, dv]
> > > > > >     """
> > > > > >
> > > > > >     # adding detach here sometimes improves stability.
> > > > > >     w0_norm = w0.norm(dim=2, keepdim=True)
> > > > > >     w1_norm = w1.norm(dim=2, keepdim=True)
> > > > > >     w2_norm = w2.norm(dim=2, keepdim=True)
> > > > > >
> > > > > >     q = q.transpose(1, 2)  # [b, dk, l]
> > > > > >     v = v.transpose(1, 2)
> > > > > >
> > > > > >     ######### update the fast weight w0, w1, w2 with test-time training #########
> > > > > >
> > > > > >     #### Forward pass with key
> > > > > >     # [b, dh, dk] @ [b, dk, l] -> [b, dh, l]
> > > > > >     gate_before_act = torch.bmm(w0, k.transpose(1, 2))
> > > > > >     hidden_before_mul = torch.bmm(w2, k.transpose(1, 2))
> > > > > >     hidden = F.silu(gate_before_act, inplace=False) * hidden_before_mul
> > > > > >
> > > > > >     #### Backward pass to compute fast weight gradients
> > > > > >     # [b, dh, dv] @ [b, dv, l] -> [b, dh, l]
> > > > > >     dhidden = torch.bmm(w1.transpose(1, 2), v)
> > > > > >
> > > > > >     dhidden_before_mul = dhidden * F.silu(gate_before_act, inplace=False)
> > > > > >     dgate = dhidden * hidden_before_mul
> > > > > >     dgate_before_act = silu_backprop(dgate, gate_before_act)
> > > > > >
> > > > > >     # [b, dv, l] @ [b, l, dh] -> [b, dv, dh]
> > > > > >     dw1 = torch.bmm(v, (hidden.transpose(1, 2) * lr1).type_as(v))  # [b, d, d]
> > > > > >     # [b, dh, l] @ [b, l, dk] -> [b, dh, dk]
> > > > > >     dw0 = torch.bmm(dgate_before_act, (k * lr0).type_as(dgate_before_act))
> > > > > >     dw2 = torch.bmm(dhidden_before_mul, (k * lr2).type_as(dhidden_before_mul))
> > > > > >
> > > > > >
> > > > > >     if use_muon:
> > > > > >         w0 = zeropower_via_newtonschulz5(dw0)
> > > > > >         w1 = zeropower_via_newtonschulz5(dw1)
> > > > > >         w2 = zeropower_via_newtonschulz5(dw2)
> > > > > >
> > > > > >     w1 = w1 + dw1
> > > > > >     w0 = w0 + dw0
> > > > > >     w2 = w2 + dw2
> > > > > >
> > > > > >     w0 = w0 / (w0.norm(dim=2, keepdim=True) + 1e-5) * w0_norm
> > > > > >     w1 = w1 / (w1.norm(dim=2, keepdim=True) + 1e-5) * w1_norm
> > > > > >     w2 = w2 / (w2.norm(dim=2, keepdim=True) + 1e-5) * w2_norm
> > > > > >
> > > > > >     ######### apply the updated fast weights to the query #########
> > > > > >
> > > > > >     # [b, dh, dk] @ [b, dk, l] -> [b, dh, l]
> > > > > >     h = torch.bmm(w2, q)
> > > > > >     gate = F.silu(torch.bmm(w0, q), inplace=True)
> > > > > >     # [b, dv, dh] @ [b, dh, l] -> [b, dv, l] -> [b, l, dv]
> > > > > >     o = torch.bmm(w1, gate * h).transpose(1, 2)
> > > > > >
> > > > > >     return o
> > > > > > ```

---

> ### Comment · Reviewer_JVw8 · 2025-08-08
> **Post reply**
>
> > "The reviewer criticized us for not being able to provide code at the time of submission."
>
> I never did criticize not providing code by itself. This is not a requirement of NeuriPS. I pointed out the inconsistency with the checklist and the fact that the implementation was being claimed as the core contribution.
>
> "Claims in the checklist appear to be poorly justified or even completely false. For instance, the authors answered [YES] to the questions regarding code sharing and reproducibility. Yet, they did not provide any code with the submission, with this claim being "justified" by a promise of sharing code conditioned on acceptance." (Original review)
>
> "I would suggest avoiding claiming that speeding up TTT N times, thanks to a new Pytorch implementation to be the paper's main contribution. I believe this is not only partially out-of-scope for a NeurIPS paper's main contribution, but it would also require at least providing reviewers with the code at the time of submission." (Reply above)
>
> Regarding your other points, I am not sure how productive this conversation is getting, as I am starting to fear that my criticism and all the other areas of improvement I highlighted in my review and prior responses are not being taken seriously, and the authors are now overly focusing on one very specific point, the code contribution of the work.

---

> > ### Author Response · Authors · 2025-08-08
> > **post-reply Sorry for the confusion.**
> >
> > Sorry about the mistake. Ok, I now understand it. It's about the inconsistency between the justification and guidelines in the checklist claim-5.
> >
> > You are right, we are wrong on claim-5. Thanks for pointing it out and clarify

---

> > > ### Author Response · Authors · 2025-08-08
> > >
> > > We appreciate the reviewer’s prompt replies and for clarifying the checklist discrepancy for multiple times.
> > >
> > > I now fully understand your point.
> > >
> > > Please let us know if you have any further comments.
> > >
> > > If there are additional concerns on other aspects of our discussion—such as our claims on speeding up the original TTT, or points 1, 5, and 6 in the “Weaknesses” section (addressed in our first-round reply)—we would be happy to respond.

---

### Official Review · Reviewer_cwFf · 2025-06-30

**Clarity:** 1
**Significance:** 2
**Originality:** 2
**Rating:** 3
**Confidence:** 3

**Summary:**

This paper addresses the computational inefficiency and scalability limitations of existing Test-Time Training (TTT) methods for long-sequence modeling. Current TTT approaches often use small mini-batches for online updates of "fast weights," leading to poor hardware utilization on modern GPUs. The authors propose Large Chunk Test-Time Training (LaCT), a method that updates fast weights using extremely large chunks of tokens, ranging from thousands to a million. To handle local dependencies within these large chunks, LaCT is combined with a window attention mechanism. The paper validates this approach across diverse modalities, including novel view synthesis, language modeling, and autoregressive video diffusion.

**Questions:**

See above

**Ethical Concerns:**

["NO or VERY MINOR ethics concerns only"]

**Final Justification:**

The authors have addressed my concerns regarding better video generation and LLM downstream task evaluations. While the method does outperform other baselines in video generation, performance on LLM related tasks are mixed.

**Limitations:**

Limitations are discussed in the Appendix only.

**Paper Formatting Concerns:**

No concerns

**Quality:**

2

**Strengths And Weaknesses:**

**Strength:**
- The LaCT approach demonstrates enhanced GPU utilization for TTT
- The authors evaluate their method on three different domains including novel view synthesis, language modeling, and autoregressive video diffusion
- Several interesting ablations (unfortunately buried in the appendix)

**Weaknesses:**
The paper's primary weaknesses lie in its empirical evaluation, unclear attribution of performance gains, and overall structure.
- **Insufficient Empirical Evaluation:** The validation of the proposed method is unconvincing across multiple domains.
   - For language modeling, the authors rely on validation loss and a simple retrieval task (S-NIAH). This is insufficient; standard practice, as seen in baselines like DeltaNet [10] that is used by the authors, involves evaluation on a suite of downstream tasks (e.g., ARC, Hellaswag) to properly assess a model's capabilities beyond simple long-context recall.
   - Similarly, for autoregressive video diffusion, performance is measured solely by denoising loss on a validation set. This metric is notoriously uncorrelated with perceptual quality. The evaluation lacks standard video generation metrics or qualitative examples, making it impossible to judge the model's actual performance in terms of temporal consistency or frame quality. The authors acknowledge this limitation in the appendix, but it remains a significant flaw in the experimental design. See the work by Yin, Tianwei, et al. [1] for relevant benchmarks and evaluation measures used in the field (e.g. temporal quality, frame quality, text alignment).

[1] Yin, Tianwei, et al. "From slow bidirectional to fast autoregressive video diffusion models." _Proceedings of the Computer Vision and Pattern Recognition Conference_. 2025.

- **Unclear Source of Performance Gains:** The paper fails to clearly disentangle the benefits of the core LaCT architecture from the optimizer used. In the language modeling experiments, the performance improvement is almost entirely dependent on the *Muon optimizer*; the variant using a momentum optimizer performs on par with the DeltaNet baseline (Figure 5). This suggests that the enhanced results may stem more from the choice of optimizer than from the proposed large-chunk training strategy itself. The authors argue that LaCT's efficiency enables the use of such sophisticated optimizers, but they do not provide evidence that other TTT methods could not also benefit from Muon, which is a critical missing ablation. Furthermore, the paper lacks direct comparisons against other relevant baselines that combine local attention with recurrence, such as InfiniAttention or MEGA, which are only briefly mentioned in the appendix's related work section.

- **Poor Paper Organization and Clarity:** The structure of the paper hinders a clear understanding of its contributions and context. A discussion of related work is relegated to the appendix, which is a significant omission for a main conference paper, especially given that Test-Time Training is not a widely studied field. This section is crucial for positioning the work and should be in the main body (related work concerning novel view synthesis and video generation can be left for the appendix). Conversely, key ablation studies that are vital to understanding the method's design choices—such as the impact of state size and different optimizers are also in the appendix. The main paper's space could have been better utilized by moving some training/implementation/dataset details to the appendix and featuring these more critical analyses instead.


Minor points:
- References 10 and 15 are the same. Only keep 15 which is a NeurIPS citation rather than arxiv.
- Figure 5: there is a typo in the word validation
- Typo in "Muon" optimizer mixed with "Moun" in several places the paper

---

> ### Author Rebuttal · Authors · 2025-07-31
>
> We thank the reviewer for their detailed feedback. We will address the main concerns below.
>
> **Reviewer comment:**
> >The authors argue that LaCT's efficiency enables the use of sophisticated optimizers but do not provide evidence that other TTT methods could not also benefit from Muon, which is a critical missing ablation.
>
> **Response:**
> We appreciate the reviewer highlighting this point. However, a direct comparison by applying sophisticated optimizers such as Muon to original test-time training methods (e.g., ttt-linear or ttt-mlp) is practically infeasible due to severe computational limitations. For instance, in our smallest language modeling experiment (760M parameters trained on 40B tokens), the original open-source ttt-linear implementation requires 32 A100 GPUs for **95 days**, and ttt-mlp requires **150 days**, whereas LaCT and other baselines complete training in approximately 20 hours on identical hardware. Integrating Muon into the original ttt-linear method would further increase the computational cost (FLOPS) by approximately 30x, making such comparisons impractical (see detailed FLOPS derivation in Equation (20) of the supplementary).
>
> Therefore, original TTT methods fundamentally cannot leverage advanced optimizers at modern scales. In contrast, our large-chunk design efficiently supports sophisticated optimizers with only dozens of lines of PyTorch code. Moreover, it enables rapid adaptation not only of test-time optimizers but also of different fast weight architectures, loss functions, and diverse data modalities efficiently as shown in the paper.
>
>
>
> **Reviewer comment:**
> >The paper fails to clearly disentangle the benefits of the core LaCT architecture from the optimizer used.
>
> **Response:**
> We respectfully disagree with the reviewer’s assessment. The primary novelty and contribution of our paper lie not simply in the choice of optimizer (e.g., Muon) but rather in the LaCT architectural framework itself. This architecture allows rapid and efficient experimentation across diverse design choices—including optimizers, fast weight architectures, loss functions, and data modalities—without relying on complex custom kernels,  extensive code modifications, or suffering from extremelly low hardware utilizations.
>
> Importantly, most deep learning researchers are unable to write cumbersome custom kernels but can readily implement PyTorch code. Thus, LaCT significantly democratizes research on test-time training and accelerates progress in next-generation efficient nonlinear recurrent neural networks.
>
> Furthermore, our paper includes extensive ablation studies clearly isolating the contributions of different design choices on both the novel view synthesis and language model experiments:
>
> - Figure 7(a): Impact of state size.
> - Figure 7(b): Effect of advanced optimizers.
> - Figure 8(a): Comparison of nonlinear versus linear fast-weight functions.
> - Figure 8(b): Evaluation of chunk recurrence versus per-token recurrence.
>
> These analyses conclusively demonstrate that nonlinear recurrent fast weights, larger state sizes, and advanced optimizers each independently enhance performance—and importantly, all these components become computationally efficient on modern GPUs only within the large-chunk architecture.
>
> We hope the above explanation clarifies our major contributions.
>
> ---
> ## On more benchmark evaluations
>
> **Reviewer comment:**
> > evaluation on a suite of downstream tasks (e.g., ARC, Hellaswag)  for language models
>
> **Response:**
> We thank the reviewer for the suggestion. We have added the suite of evaluations:
>
> | Model | ARC-c (acc) | ARC-e (acc) | Hella. (acc_norm) | PIQA (acc_norm) | BoolQ (acc) | Wino (acc) | OpenBook (acc) | LAMBADA (acc) | SciQ (acc) | LAMBADA ppl ↓ | Wiki ppl ↓ | Avg. (acc) ↑ |
> |:---|---:|---:|---:|---:|---:|---:|---:|---:|---:|---:|---:|---:|
> | **760M models** | | | | | | | | | | | | |
> | Transformer 760M | 0.247 | 0.465 | 0.405 | 0.664 | 0.588 | 0.526 | 0.190 | 0.375 | 0.864 | 22.574 | 20.750 | **0.496** |
> | Transformer SWA 760M | 0.234 | 0.479 | 0.418 | 0.663 | 0.456 | 0.502 | 0.172 | 0.407 | 0.879 | 19.537 | 21.215 | **0.496** |
> | Ours Momentum | 0.247 | 0.474 | 0.413 | 0.645 | 0.611 | 0.523 | 0.188 | 0.393 | 0.861 | 20.519 | 21.300 | **0.500** |
> | Ours Muon | 0.235 | 0.477 | 0.420 | 0.653 | 0.502 | 0.534 | 0.182 | 0.392 | 0.897 | 20.584 | 20.702 | **0.496** |
> | GLA SWA | 0.238 | 0.475 | 0.420 | 0.649 | 0.501 | 0.528 | 0.184 | 0.393 | 0.897 | 20.621 | 20.695 | **0.502** |
> | DeltaNet SWA | 0.253 | 0.466 | 0.417 | 0.650 | 0.608 | 0.515 | 0.184 | 0.386 | 0.888 | 20.415 | 23.285 | **0.502** |
> | **3B+ models** | | | | | | | | | | | | |
> | Transformer | 0.258 | 0.488 | 0.515 | 0.655 | 0.548 | 0.539 | 0.204 | 0.479 | 0.900 | 12.092 | 15.570 | **0.517** |
> | Transformer SWA | 0.243 | 0.462 | 0.522 | 0.629 | 0.557 | 0.554 | 0.214 | 0.490 | 0.905 | 11.751 | 15.737 | **0.513** |
> | Ours Momentum | 0.273 | 0.496 | 0.529 | 0.628 | 0.556 | 0.533 | 0.218 | 0.480 | 0.905 | 11.419 | 15.439 | **0.528** |
> | Ours Muon | 0.263 | 0.473 | 0.523 | 0.621 | 0.522 | 0.546 | 0.210 | 0.495 | 0.894 | 10.988 | 15.313 | **0.516** |
> | GLA SWA | 0.259 | 0.466 | 0.517 | 0.628 | 0.508 | 0.535 | 0.190 | 0.465 | 0.897 | 12.547 | 17.284 | **0.506** |
> | DeltaNet SWA | 0.255 | 0.475 | 0.504 | 0.648 | 0.491 | 0.547 | 0.190 | 0.453 | 0.891 | 12.682 | 17.950 | **0.498** |
>
>
>
> As the reviewer noted, these benchmarks primarily assess the model's general commonsense knowledge. However, these metrics alone do not precisely capture the long-context modeling capability of the models. In contrast, the per-position-loss and retrieval task (S-NIAH), used in our original evaluation, specifically measure the model's ability to handle and retrieve information from longer contexts.
>
> **Reviewer comment:**
> > lacks standard video generation metrics.
>
> **Response:**
> We follow Yin et al[1]. to include full suite of metrics in Vbench[2]. This concern is also raised by reviewer 3KQP.
>
> Due to character limits, we put the table of VBench results in replies of reviewer 3KQP.
>
> [1] Yin, Tianwei, et al. "From slow bidirectional to fast autoregressive video diffusion models." CVPR2025
>
> [2] VBench: Comprehensive Benchmark Suite for Video Generative Models. CVPR 2024.
>
> ---
> ## On Comparison with Additional Baselines
>
> **Reviewer comment:**
> >The paper lacks direct comparisons against other relevant baselines that combine local attention with recurrence, such as InfiniAttention or MEGA.
>
> **Response:**
> In our language modeling experiments, the main paper already compares our method against two state-of-the-art baselines combining local attention with advanced recurrence: GLA with SWA and DeltaNet with SWA.
>
> Here, we additionally provide a direct comparison with InfiniAttention. Since there is no official open-source implementation available, we implemented InfiniAttention ourselves following Equations (7)-(10) from their paper. Our initial implementation encountered gradient explosion issues after approximately 20K iterations. To enhance stability, we added two additional normalization layers:
> - An RMS normalization layer on the output of the memory module, as is done in GLA, DeltaNet and LaCT
> - Weight normalization on the linear recurrent memory as is done in LaCT.
>
> Both normalization methods significantly improved stability for InfiniAttention.
>
> We evaluate the models using the average per-position loss over 2.5 billion validation tokens, consistent with Figures 5(a) and 5(c). Due to rebuttal limitations on adding new figures, we present the results in the table below, showing the average token loss for each 4K-token chunk across eight chunks (total sequence length of 32,768 tokens):
>
> | Method | Chunk 1 | Chunk 2 | Chunk 3 | Chunk 4 | Chunk 5 | Chunk 6 | Chunk 7 | Chunk 8 |
> |--------|---------|---------|---------|---------|---------|---------|---------|---------|
> | Ours Momentum | 2.6458 | 2.5845 | 2.5777 | 2.5745 | 2.5719 | 2.5716 | 2.5703 | 2.5692 |
> | Infini-Attention | 2.8635 | 2.8362 | 2.8369 | 2.8370 | 2.8363 | 2.8370 | 2.8365 | 2.8355 |
>
> We also evaluate on S-NIAH:
>
> | Model | S-NIAH-1 4K | S-NIAH-1 8K | S-NIAH-1 16K | S-NIAH-2 4K | S-NIAH-2 8K | S-NIAH-2 16K | S-NIAH-3 4K | S-NIAH-3 8K | S-NIAH-3 16K |
> | :--- | :---: | :---: | :---: | :---: | :---: | :---: | :---: | :---: | :---: |
> | Ours Momentum | 95.6 | 84.8 | 83.4 | 91.4 | 73.4 | 22.8 | 82.6 | 34.8 | 16.6 |
> | Infini-Attention | 45 | 18.2 | 8.2 | 38 | 25.6 | 12.2 | 12 | 15.8 | 7.2 |
>
> We have two intuitions about InfiniAttention's lower performance:
>
> - It employs chunked window attention rather than sliding window attention, limiting its ability to effectively capture continuous context.
> - Its recurrence uses a chunk-level linear recurrence with a delta rule, which typically has lower expressiveness compared to the per-token linear recurrence with delta-rule used by DeltaNet in language task.
>
>
>
> ### On Paper Organization and Clarity
>
> We thank the reviewer for the excellent suggestions on improving the paper's structure. We agree and will make the following changes in the revised version:
>
> - Related Work: The core "Related Work" section on Test-Time Training will be moved from the appendix to the main body of the paper to better position our contributions.
>
> - Ablation Studies: The key ablation studies (Figures 7 and 8), which are vital for understanding the method's design, will be moved into the main paper.
>
> - Experimental Details: To create space for these additions, less critical implementation and dataset details from Section 5 will be moved to the appendix.
>
>
> - Minor Corrections
> We thank the reviewer for catching these errors. We will correct the duplicate reference and fix all typos (including "validation" and "Muon") in the final version.

---

> > ### Comment · Reviewer_cwFf · 2025-08-06
> >
> > I would like to thank the authors for their comprehensive response. Thank you for adding the new benchmark results on Video Gen and downstream LLM tasks, as well as comparison to Infini-Attention.
> >
> > The authors mentioned:
> > > Our method consistently performs on par or better than other autoregressive video generation baselines, regarding temporal and frame quality.
> >
> > Could you please elaborate your finding on VBench results. I see that the "Frame Quality" and "Text Alignment" have deteriorated in performance, right?
> >
> > For full comparison the authors should report the `Total Score` as can be found in the leaderboard. Can the authors report the total score? "https://huggingface.co/spaces/Vchitect/VBench_Leaderboard"
> >
> > Also could you clarify why the overall consistency score of the Original Wan (14B) model 0.2499 is much lower than the results in the leaderboard 0.2749?

---

> ### Author Response · Authors · 2025-08-06
> **Round-2 response to reviewer cwFf**
>
> I appreciate the reviewer’s engagement. Here we address your new comments:
> ---
>
> Reviewer's comment
> > Could you please elaborate your finding on VBench results. I see that the "Frame Quality" and "Text Alignment" have deteriorated in performance, right?
> >
> Response:
>
> When comparing our method (1.3B) against Transformer SWA (1.3B) and Transformer (1.3B), our approach achieves higher scores in both `Temporal Quality` (0.9250 vs. 0.9114/0.9232) and `Frame Quality` (0.6360 vs. 0.6303/0.6219). However, there is a minor decrease in `Text Alignment` (0.2475 vs. 0.2477/0.2478), indicating a slight trade-off.
>
> Reviewer's comment
> > Report the Total Score
>
> Response:
>
> Thank you for this suggestion. We have computed and provided the Total Score using the official VBench scripts, which aggregate and normalize all individual scores. Here are the results:
>
>
> | Type | Method | **Temporal&nbsp;Quality** | **Frame&nbsp;Quality** | **Text&nbsp;Alignment** | **Quality&nbsp;Score** | **Semantic&nbsp;Score** | **Total&nbsp;Score** |
> | :--- | :--- | :---: | :---: | :---: | :---: | :---: | :---: |
> | Bidirectional | Original Wan (1.3 B) | 0.9194 | 0.6427 | 0.2506 | 0.8359 | 0.6713 | 0.8030 |
> | AR | Ours (1.3 B) | 0.9250 | 0.6360 | 0.2475 | 0.8251 | 0.6217 | 0.7844 |
> | AR | Transformer SWA (1.3 B) | 0.9114 | 0.6303 | 0.2477 | 0.8162 | 0.6009 | 0.7731 |
> | AR | Transformer (1.3 B) | 0.9232 | 0.6219 | 0.2478 | 0.8228 | 0.6040 | 0.7790 |
> | **14 B parameters** |  |  |  |  |  |  |  |
> | Bidirectional | Original Wan (14 B) | 0.9308 | 0.6452 | 0.2579 | 0.8431 | 0.6953 | 0.8135 |
> | AR | Ours (14 B) | 0.9279 | 0.6330 | 0.2567 | 0.8289 | 0.6586 | 0.7949 |
>
>
>
> Reviewer's comments
> > Also could you clarify why the overall consistency score of the Original Wan (14B) model 0.2499 is much lower than the results in the leaderboard 0.2749?
> >
>
> Response:
>
> Thanks for pointing out this discrepancy. Upon investigating this inconsistency, I found a related github issue (#163, "Inquiry about Wan2.1-T2V-14B Results on vbench (2025-07-08)") of the VBench GitHub repository.  (We apologize for not being able to include direct GitHub issue links during the rebuttal period.)
>
> According to the latest response in that issue from last week, the Original Wan model on the leaderboard was evaluated using the augmented prompts with longer text description. (my guess is that the Wan T2V API would rewrites short captions into more detailed text prompts that align closely with their training caption distribution.).  Also, the leaderboard results sampled the videos for five times, and we sampled each prompt one time.
>
> We can regenerate videos using these augmented text prompts and re-running evaluations. We will update our results accordingly if we can complete this task by August 8th (the end of the author response period).

---

> ### Comment · Reviewer_cwFf · 2025-08-08
>
> Thank you for your clarification and reporting the total score. I believe the strongest results are obtained for this video generation task. The gains for the LLM related tasks seem insignificant or slightly worse for 760M, but still informative to report.
>
> Overall, the new results have made the evaluations more convincing, and I will raise the rating to 3. I will engage in the reviewer's discussion and may increase the score further.

---

> > ### Author Response · Authors · 2025-08-09
> > **Round-3 response to reviewer cwFf**
> >
> > We appreciate the reviewer’s engagement during the discussion stage and their helpful feedback in improving our paper.
> >
> > We now provide new evaluation results on VBench using the WAN official augmented prompt set. Compared to the original VBench prompt set—which contains shorter, less detailed descriptions—all six major scores improve when using augmented prompts.
> >
> > Consistent with the observation of using the standard vbench prompt set, our method continues to outperform autoregressive baselines in-terms of `temporal quality` , `semantic score` and  `total score` , with other metrics remaining on par, though the performance gap narrows when using augmented text captions.
> >
> > The results for Wan-14B (Bidirectional) and Ours-14B (Autoregressive) are still being generated and will be included in the revised paper.
> >
> >
> > | Type          | Method                 | Temporal Quality | Frame Quality | Text Alignment | Quality Score | Semantic Score | Total Score |
> > |---------------|------------------------|------------------|---------------|----------------|---------------|----------------|-------------|
> > | Bidirectional | Original Wan (1.3 B)   | 0.9220           | 0.6652        | 0.2616         | 0.8448        | 0.7598         | 0.8278      |
> > | AR            | Ours (1.3 B)           | 0.9397           | 0.6658        | 0.2604         | 0.8441        | 0.7425         | 0.8238      |
> > | AR            | Transformer SWA (1.3 B)| 0.9320           | 0.6653        | 0.2609         | 0.8448        | 0.7300         | 0.8218      |
> > | AR            | Transformer (1.3 B)    | 0.9362           | 0.6637        | 0.2600         | 0.8457        | 0.7295         | 0.8225      |
> >
> > ---
> >
> > Detailed Subscores on VBench using WAN augmented prompt set.
> >
> > | Type | Method | Subject Consistency | Background Consistency | Temporal Flickering | Motion Smoothness | Dynamic Degree | Aesthetic Quality | Imaging Quality | Object Class | Multiple Objects | Human Action | Color | Spatial Relationship | Scene | Temporal Style | Appearance Style | Overall Consistency |
> > |---|---|---:|---:|---:|---:|---:|---:|---:|---:|---:|---:|---:|---:|---:|---:|---:|---:|
> > | Bidirectional | Original Wan (1.3B) | 0.9487 | 0.9664 | 0.9931 | 0.9855 | 0.6528 | 0.6571 | 0.6734 | 0.8722 | 0.7321 | 0.9500 | 0.9059 | 0.7453 | 0.4509 | 0.2302 | 0.2158 | 0.2536 |
> > | AR | Ours (1.3B) | 0.9287 | 0.9477 | 0.9749 | 0.9819 | 0.8611 | 0.6536 | 0.6781 | 0.8813 | 0.6303 | 0.9200 | 0.8983 | 0.7095 | 0.4709 | 0.2270 | 0.2135 | 0.2545 |
> > | AR | Transformer SWA (1.3B) | 0.9169 | 0.9463 | 0.9875 | 0.9822 | 0.8333 | 0.6480 | 0.6826 | 0.8837 | 0.6265 | 0.9500 | 0.8624 | 0.6675 | 0.4179 | 0.2272 | 0.2131 | 0.2556 |
> > | AR | Transformer (1.3B) | 0.9226 | 0.9468 | 0.9839 | 0.9835 | 0.8472 | 0.6472 | 0.6802 | 0.8204 | 0.6623 | 0.9600 | 0.8258 | 0.7245 | 0.4070 | 0.2293 | 0.2107 | 0.2587 |
> >
> > ---
> >
> > Reviewer:
> > > "The gains for the LLM related tasks seem insignificant or slightly worse for 760M, but still informative to report."
> >
> > We agree that including these general-capability LLM benchmark is informative. Together with the per-position-loss and S-NIAH results already in the paper—which focus on long-context performance—this addition makes the evaluation more comprehensive.

---

### Official Review · Reviewer_3KQP · 2025-07-07

**Clarity:** 3
**Significance:** 4
**Originality:** 3
**Rating:** 5
**Confidence:** 3

**Summary:**

This paper presents Large Chunk Test-Time Training (LaCT), a new framework for test-time training (TTT) that significantly improves hardware efficiency and scalability for long-context modeling. Traditional TTT approaches rely on small mini-batch updates (e.g., every 16 or 64 tokens), leading to poor FLOPs utilization (<5%) and limited fast-weight state capacity. LaCT instead proposes using large chunks of data—ranging from 2K to over 1M tokens—for fast-weight updates, enabling 10–15× higher utilization (up to 70% on A100 GPUs), and scaling state size up to 40% of total model parameters.

LaCT is evaluated on three diverse tasks: (1) novel view synthesis with up to 1 million token context from 128 input views; (2) long-context language modeling (up to 32K context length) with 760M and 3B parameter models; and (3) autoregressive video diffusion using a 14B parameter model. Across all domains, LaCT achieves comparable or superior performance to state-of-the-art baselines (e.g., DeltaNet, Mamba, 3D Gaussian Splatting), while maintaining high throughput and implementation simplicity using native PyTorch code.

**Questions:**

N/A

**Ethical Concerns:**

["NO or VERY MINOR ethics concerns only"]

**Final Justification:**

I keep my original score, and think this paper should get accepted

**Quality:**

3

**Strengths And Weaknesses:**

### strength

1.Hardware-efficient design through large chunk updates:
The key idea of large-chunk updates (e.g., 2K–1M tokens) allows significant FLOPs utilization improvements. For example, on novel view synthesis with 196K input tokens, LaCT achieves 1.4s prefill time and 38.7 FPS rendering, while Perceiver and full attention baselines take over 16s prefill and <35 FPS (Table 2).

2.Effective across modalities with flexible architecture:
LaCT is applied to image sets (e.g., Google Scanned Objects and DL3DV datasets), text (Long-Data-Collections), and video (autoregressive denoising in Wan 2.1). In language modeling, a 3B LaCT model with Muon optimizer achieves lower per-token loss and higher retrieval accuracy (S-NIAH-2) than GLA and DeltaNet baselines (Figure 5c–d).

3.Nonlinear fast weights and advanced update rules:
The paper shows that using SwiGLU MLPs for fast weights, combined with the Muon update rule, yields strong empirical gains. For instance, in LM tasks, the Muon variant consistently outperforms Momentum and vanilla gradient descent (Figure 7b).

4.Modular, scalable, and easily implemented:
LaCT’s design—chunked memory updates + local window attention—enables native PyTorch implementation in “a few dozen lines of code” while supporting distributed context-parallelism. On large-scale video generation (14B params), it autoregressively generates sequences of up to 56K tokens with stable training and clean decoding (Figure 6c).

### weakness
1. The paper uses denoising loss at multiple timesteps as its primary metric for autoregressive video. However, no perceptual or user-based evaluation (e.g., FVD, CLIP-similarity, human rating) is provided, making it hard to assess real-world generative quality.
2. Why the model learning has a big performance gap when using momentum or moun based optimizer for LM (figure 5)

---

> ### Author Rebuttal · Authors · 2025-07-31
>
> We thank the reviewer for their constructive and insightful feedback. Below we address the key questions and suggestions:
>
>
> ## VBench Results
> > More perceptual evaluations for video generations
>
> **Response:**
> We appreciate the reviewer's suggestion, which also aligns with Reviewer cwFf’s comment. To provide more comprehensive perceptual evaluations, we conducted additional experiments using the widely recognized VBench benchmark suite [2], following evaluation practices established by Yin et al. [1].
>
> Specifically, we used 942 standard text prompts from VBench and generate videos for all compared models.  For our 1.3B-parameter model, we generated two videos per prompt. For our 14B-parameter model, we generated one video per prompt due to time constraints.
>
> Below are the summarized evaluation results using standard VBench metrics, which we will include in the revised manuscript:
>
>
> | Type | Method | **Temporal Quality** | **Frame Quality** | **Text Alignment** |
> | :--- | :--- | :---: | :---: | :---: |
> | Bidirectional | Original Wan (1.3B) | 0.9194 | 0.6427 | 0.2506 |
> | AR | Ours (1.3B) | 0.9250 | 0.6360 | 0.2475 |
> | AR | Transformer SWA (1.3B) | 0.9114 | 0.6303 | 0.2477 |
> | AR | Transformer (1.3B) | 0.9232 | 0.6219 | 0.2478 |
> |14B parameters|:---|:---:|:---:|:---:|
> | Bidirectional | Original Wan (14B) | 0.9308 | 0.6452 | 0.2579 |
> | AR | Ours (14B) | 0.9279 | 0.6330 | 0.2567 |
>
> ***
>
> Our method consistently performs on par or better than other autoregressive video generation baselines, regarding temporal and frame quality.
>
> Below is the detailed metric comparison table (we would include a radar chart version in the revised version):
>
> | Type | Method | Subject Consistency | Background Consistency | Temporal Flickering | Motion Smoothness | Dynamic Degree | Aesthetic Quality | Imaging Quality | Object Class | Multiple Objects | Human Action | Color | Spatial Relationship | Scene | Temporal Style | Appearance Style | Overall Consistency |
> | :--- | :--- | :---: | :---: | :---: | :---: | :---: | :---: | :---: | :---: | :---: | :---: | :---: | :---: | :---: | :---: | :---: | :---: |
> | Bidirectional | Original Wan (1.3B) | 0.9536 | 0.9660 | 0.9942 | 0.9824 | 0.6319 | 0.6110 | 0.6744 | 0.7698 | 0.6086 | 0.7750 | 0.9191 | 0.7269 | 0.1944 | 0.2362 | 0.2024 | 0.2364 | 0.9194 | 0.6427 | 0.2506 |
> | AR | Ours (1.3B) | 0.9394 | 0.9447 | 0.9698 | 0.9820 | 0.7431 | 0.5981 | 0.6740 | 0.6966 | 0.4474 | 0.7350 | 0.8139 | 0.6601 | 0.2169 | 0.2312 | 0.2041 | 0.2292 |
> | AR | Transformer SWA (1.3B) | 0.9223 | 0.9149 | 0.9762 | 0.9811 | 0.7431 | 0.5859 | 0.6746 | 0.6610 | 0.3575 | 0.7550 | 0.8149 | 0.5891 | 0.2118 | 0.2314 | 0.2026 | 0.2286 |
> | AR | Transformer (1.3B) | 0.9290 | 0.9368 | 0.9805 | 0.9800 | 0.7708 | 0.5836 | 0.6602 | 0.6776 | 0.3708 | 0.7000 | 0.8517 | 0.6199 | 0.1973 | 0.2306 | 0.2022 | 0.2312 |
> | Bidirectional | Original Wan (14B) | 0.9504 | 0.9689 | 0.9928 | 0.9844 | 0.7083 | 0.6227 | 0.6678 | 0.8196 | 0.6280 | 0.7700 | 0.9024 | 0.7296 | 0.2733 | 0.2338 | 0.2136 | 0.2499 |
> | AR | Ours (14B) | 0.9357 | 0.9469 | 0.9765 | 0.9831 | 0.7639 | 0.5983 | 0.6678 | 0.7326 | 0.4985 | 0.8500 | 0.8618 | 0.6362 | 0.2224 | 0.2328 | 0.2186 | 0.2451 |
>
> [1] Yin, T., et al. "From slow bidirectional to fast autoregressive video diffusion models." CVPR 2025.
>
> [2] VBench: Comprehensive Benchmark Suite for Video Generative Models, CVPR 2024.
>
>
> ## Understanding Muon
>
> > Why the model learning has a big performance gap when using momentum or moun based optimizer for LM (figure 5)
>
> **Response:**
> This is an insightful question, and we have some initial intuitions.
>
> First, some background: Muon is a novel optimizer recently adopted by leading AI labs as a replacement for AdamW in large language model pretraining (see Kimi K2 [3]). In our paper, we demonstrate the effectiveness of Muon not during pretraining but specifically during test-time training, as illustrated in Figures 5 and 7(b) on page 15.
>
> Understanding why Muon performs effectively is an active area of ongoing research. One recent paper that significantly influenced our thinking is reference [4], which shows that spectral normalization methods on gradients like Muon can notably enhance model learning under extreme label imbalance conditions.
>
> This intuition aligns closely with our S-NIAH results, where Muon considerably improves performance (Figure 5(b, d)). Specifically, when evaluating on S-NIAH, the model must memorize small segments of passkeys embedded within irrelevant contexts, analogous to label imbalance scenarios. Because Muon orthogonalizes and normalize the spectral of gradients, it amplifies gradient magnitudes associated with less frequent tokens, thereby facilitating the model's ability to retain long-tail information.
>
> These insights are preliminary hypotheses and may require further validation.  We intend to add this point in an appendix paragraph for revised version. We hope this perspective inspires both reviewers and future readers.
>
> [3] Kimi K2: Open Agentic Intelligence. arxiv 2025.
>
> [4] On Generalization of Spectral Gradient Descent: A Case Study on Imbalanced Data

---

### Author Response · Authors · 2025-08-09
**Author final response**

We sincerely thank all reviewers for their constructive feedback and engagement, and greatly appreciate your efforts in helping us strengthen this work!

---

### Decision · Program_Chairs · 2025-09-17

**Decision:**

Reject

**Comment:**

The paper introduces LaCT, a framework for efficient test-time training (TTT) that aims to improve hardware utilization and scalability for long-context modeling. Reviewers 3KQP and Zdxy highlight the method’s generality, hardware efficiency, and ease of implementation, as well as its empirical results in view synthesis and video generation. Reviewer cwFf notes the improved GPU utilization and the breadth of evaluation, and appreciates the comprehensive ablations.
However, significant concerns were raised in the reviews, especially by reviewers cwFf and JVw8. There are major concersn on regarding the empirical evaluation, clarity of attribution for performance gains, and novelty of the work. For language experiments, the evaluation was limited to validation loss and a simple retrieval task, lacking standard downstream benchmarks. For video diffusion, the initial evaluation relied the loss, which is not well correlated with perceptual quality; standard metrics and qualitative results were missing in the original submission. Reviewer JVw8 was also concerned that the core contribution may be limited to implementation and hyperparameter changes, and that the claims about hardware efficiency are too dependent on specific hardware and not sufficiently contextualized. Both JVw8 and cwFf also note that the paper’s organization/presentation can be improved. The AC read the paper and agrees with the potential to improve clarity.

In terms of ratings , the four reviewers recommend the following:
3KQP maintained an accept rating after rebuttal, mentioning technical solidity and expressing satisfaction with the authors’ additional evaluations and clarifications. The AC believes that the strengths in this review are not particularly well contextualized within the research community, but rater a restatement of the paper's contributions.
cwFf initially raised concerns about evaluation and clarity, but after the authors provided new results on VBench and LLM downstream tasks, as well as a comparison to InfiniAttention, acknowledged that the video generation results are strong and the new evaluations are more convincing. The reviewer still maintained their borderline reject rating noting that LLM results are mixed but informative.
JVw8 kept a negative assessment, arguing that the main contribution is an efficient PyTorch implementation rather than a fundamentally new method, and that the claims about speedup and code sharing were not fully justified. The reviewer did not change their rating after extensive discussion, despite the authors’ clarifications and code snippets.
Zdxy is consistently positive, highlighting the generality, and results, recommending acceptance, but nothing very concrete (e.g. the strengths mentioned in this review are rather shallow).

Given the split in reviewer opinion, the questions around clear and substantial novelty (much of the improvement appears to stem from implementation and hardware-specific optimizations rather than a fundamentally new algorithmic idea), and the concerns about evaluation (especially for language modeling), clarity, and reproducibility, the AC believes this paper is not ready for publication at NeurIPS. The paper would benefit from a more thorough evaluation, clearer attribution of performance gains, and a more clear framing of the contributions and respective ablations to show their value. Despite the authors’ responsiveness during the rebuttal, the paper’s main weaknesses remain.The overstatement of some claims regarding reproducibility and speedup further undermine confidence in the work (note, this is not a sole reason for rejection). The authors are encouraged to address all the issues raised by the reviewers when preparing a revised version.